# CRED: Contrastive Residual Embedding Decoding for Adaptive Concept Unlearning

## Abstract

Large language models (LLMs) trained on web-scale data inevitably encode outdated, private, or undesired knowledge, posing challenges for privacy, safety, and factual reliability. Existing machine unlearning methods typically require retraining or finetuning to remove targeted knowledge, which is costly and prone to catastrophic forgetting of useful capabilities. In this work, we propose **Contrastive Residual Embedding for Decoding (CRED)**, an in-context unlearning method that operates entirely at inference time with no parameter updates. Given a query, CRED constructs retrieval-augmented prompts from a retain set and a forget set, computes a contrastive residual from multi-layer decoder embeddings, and injects this residual into the decoder of the original prompt. The residual steers generation away from forget set content while preserving knowledge supported by the retain set. Experiments on the TOFU and MUSE benchmarks demonstrate that CRED achieves effective concept erasure with minimal quality degradation. We further evaluate deployment readiness with 8/4-bit inference and summarize stability via a simple quantization deviation metric. CRED maintains consistent forgetting and utility across compression and combines easily with retrieval, providing a training-free path to auditable unlearning at inference time.

## 1 Introduction

Machine unlearning has become a critical tool for maintaining the reliability and safety of large language models (LLMs). As LLMs are increasingly deployed in sensitive domains such as healthcare, law, and finance, it is essential to suppress outdated, incorrect, or sensitive information selectively. Practical applications of unlearning include regulatory compliance (*e.g.*, GDPR (Wang et al., 2025a)), mitigation of misinformation and harmful content, and adaptation to evolving data. Unlearning also enables the upkeep of organizational knowledge bases by removing deprecated procedures or confidential materials that should no longer influence downstream use (Chen & Yang, 2023; Yao et al., 2024a; Foster et al., 2024).

Current approaches to machine unlearning generally involve fine-tuning or directly editing the model parameters to enforce forgetting specific concepts (Zheng et al., 2023a; Jia et al., 2024; Zhang et al., 2024b). These methods include techniques like knowledge editing, weight pruning, and adversarial training. While effective, these approaches suffer from substantial drawbacks. For example, each new piece of information identified for removal requires an expensive retraining or fine-tuning process. Such retraining-based approaches require considerable computational resources, which limits their scalability and practical deployment in rapidly evolving scenarios. Moreover, repeated parameter adjustments can lead to catastrophic forgetting and inadvertently degrade the model's performance and utility on previously learned information. Ensuring the preservation of relevant, unaffected knowledge during repeated retraining processes poses a significant challenge, often requiring meticulous monitoring and additional optimization strategies.

To address these limitations, retrieval-augmented generation (RAG) has been explored as a promising inference-time approach, allowing dynamic control over generated content without parameter updates (Fan et al., 2024). A recent work demonstrated the potential of using RAG for unlearning (Wang et al., 2025b). However, this work primarily frames the problem as a safety alignment task, often defaulting to non-informative responses such as "sorry" or disclaimers when encountering sensitive queries. This conservative approach limits the practical utility of the model, as it

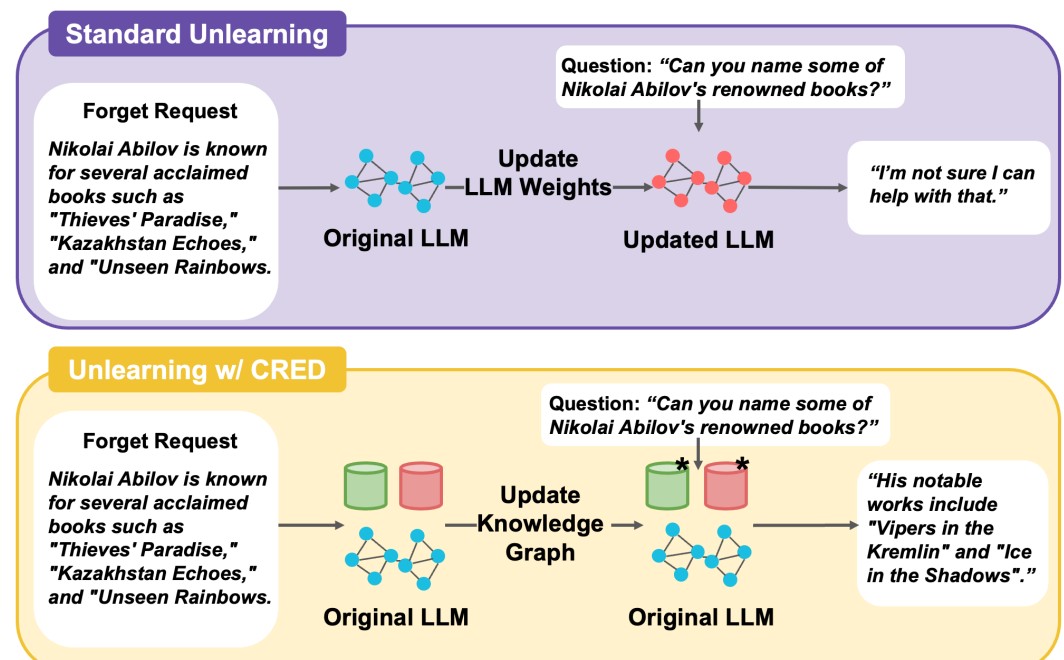

Figure 1: **Comparison of standard retraining-based unlearning (top) and the proposed CRED method (bottom).** Standard methods achieve unlearning by updating model parameters, often resulting in vague or evasive responses. In contrast, CRED performs inference-time unlearning without modifying the model. It uses contrastive decoding guided by a knowledge graph to selectively suppress or retain information.

unnecessarily restricts responses even when ample non-sensitive information remains available. For example, when queried about personal details, a model should ideally refrain from providing sensitive specifics like a phone number or home address but could still comfortably disclose less sensitive information, such as affiliated universities or published research. Another instance would be queries related to copyrighted content; rather than providing no information at all, a model could provide general descriptive insights or contextually relevant, non-infringing information.

To address these limitations, we propose **Contrastive Residual Embedding for Decoding (CRED)**, an inference-time machine unlearning method that does not require any parameter updates to the language model. CRED leverages differences between retrieval-augmented generation outputs conditioned on retain and forget knowledge graphs. For each query, the method constructs three types of prompts: an original prompt, a retain prompt based on the retain knowledge graph, and a forget prompt based on the forget knowledge graph. By comparing decoder logits under the retain and forget conditions, CRED computes a contrastive residual vector that steers the model's output away from unwanted concepts during token generation. To ensure effective and stable unlearning, the method applies a layer-wise fusion mechanism that combines adjusted logits from multiple decoder layers, thereby preserving semantic coherence and fluency.

Figure 1 contrasts parameter-editing approaches with our method. Training-based unlearning typically requires updating model weights and often yields vague or evasive responses. By contrast, CRED operates entirely at inference time: it updates only the external knowledge graph while leaving the language model unchanged, thereby suppressing the specified knowledge while preserving informativeness and contextual appropriateness. The method is lightweight, modular, and readily composable with existing inference pipelines.

Practical deployments often rely on post-training quantization to meet efficiency constraints, but prior work has shown that quantization can undermine unlearning by restoring forgotten knowledge at low precision (Zhang et al., 2025). To assess this risk, we introduce a quantization deviation measure that captures differences from both the retain baseline and the full-precision model. Results on

MUSE-News (Figure 3, Section 5.3) show that CRED achieves the best stability, with the lowest deviation across metrics, thereby preserving unlearning effectiveness and utility even under aggressive compression.

Our key contributions are as follows:

- **Limitations of prior work.** We identify significant limitations of existing training-based and inference-time unlearning methods, particularly their computational inefficiency, limited flexibility, and overly conservative handling of sensitive information.

- **Inference-time contrastive unlearning.** We introduce an innovative inference-time, retrieval-augmented unlearning method that leverages contrastive decoding to dynamically and effectively suppress undesired concepts without retraining, significantly reducing computational overhead.

- **Effectiveness.** CRED outperforms state-of-the-art baselines: on TOFU it delivers more than 3x higher forget quality without sacrificing model utility; on MUSE it meets all three criteria (low memorization on the forget set and high memorization on the retain set) with competitive privacy leakage, yielding a favorable safety–utility trade-off.

- **Quantization robustness.** We evaluate robustness under 8-bit and 4-bit inference using a simple quantization deviation measure—the sum of absolute differences across precisions. CRED achieves the lowest deviation across metrics, demonstrating the best stability under compression while preserving unlearning and utility, making it well suited for efficient deployment.

## 2 RELATED WORK

**Unlearning via Further Training.** Early machine-unlearning approaches typically require direct access to model parameters and training processes (Cao & Yang, 2015; Ginart et al., 2019; Golatkar et al., 2020; Guo et al., 2020). Gradient-based techniques, including GRADASCENT, GRADDIFF, MISMATCH, FLAT, and LLMU, modify model parameters by anti-training or optimizing contrastive objectives that efficiently handle large-scale deletions (Maini et al., 2024; Liu et al., 2022; Wang et al., 2025c; Liu et al., 2024; Yao et al., 2024b). Preference-based methods such as PO, DPO, and NPO frame unlearning as preference optimization, fine-tuning models to favor retained information and suppress unwanted knowledge (Maini et al., 2024; Rafailov et al., 2023; Zhang et al., 2024a). However, these methods are computationally intensive and require retraining whenever the retained or forgotten knowledge changes. This motivates the development of alternative approaches that operate at inference time without full retraining.

**Inference-Time Unlearning.** Training-free methods address unlearning without modifying model parameters. Simple approaches, such as output filtering, mask undesired tokens but fail to provide meaningful alternatives (Thaker et al., 2024). More sophisticated methods include In-Context Unlearning (ICUL), which explicitly inserts forget examples into prompts to induce unlearning (Pawelczyk et al., 2024), and GUARD, which penalizes the generation of banned phrases (Deng et al., 2025). Contrastive decoding was originally proposed to improve text generation quality by contrasting a strong "expert" model with a weaker "amateur" model (Li et al., 2023); it has since been adapted for toxicity mitigation, factual consistency, and controllable generation (Niu et al., 2024; Lv et al., 2024; Zheng et al., 2023b). UCD applies contrastive decoding specifically to unlearning. It preserves the original LLM, trains two small auxiliaries on the retain and forget sets, and adjusts the logits at each decoding step by subtracting their difference (Suriyakumar et al., 2025). In contrast, our CRED operates differently in two key ways. It contrasts contexts rather than models by running a single black-box model twice—once with retain evidence and once with forget evidence—and it grounds these contexts in KG-guided retrieval, enabling suppression of open-ended factual content without auxiliary training. Unlike other inference-time or retrieval-based methods (Shi et al., 2024; Wang et al., 2023; Liang et al., 2025), this design allows dynamic suppression of unwanted information without parameter updates or offline corpus modifications.

## 3 Preliminary

This section states the foundations of our setting. We first formalize machine unlearning as a partition of a knowledge corpus into retain and forget subsets. We then describe how to construct a knowledge graph from textual chunks that couples symbolic triples with semantic embeddings. Finally, we present a graph-guided retrieval routine that encodes queries, performs multi-hop matching, and ranks candidate chunks, which will provide structured evidence for the contrastive decoding stage.

**Problem Formulation.** Machine unlearning removes the influence of specified knowledge while preserving overall utility. Let $\mathcal{D}$ be a text corpus partitioned into two disjoint sets: a *retain* set $\mathcal{D}_r$ and a *forget* set $\mathcal{D}_f$, with $\mathcal{D}_r \cap \mathcal{D}_f = \emptyset$. Given a query $x$, the model should avoid content grounded in $\mathcal{D}_f$ yet remain fluent and informative based on $\mathcal{D}_r$. We adopt the *inference-time* paradigm: forgetting is achieved during decoding without updating model parameters.

**Knowledge Graph Construction.** We build a knowledge graph $\mathcal{G}$ from the corpus $\mathcal{D}$ in three steps. First, split $\mathcal{D}$ into text chunks $\{c\}$. Let $\mathcal{E}$ denote the set of entities and $\mathcal{R}$ the set of relations. We apply a triple extractor $T_\phi : \text{text} \to \mathcal{E} \times \mathcal{R} \times \mathcal{E}$ to obtain factual triples $(h, r, t)$ for each $c$, where $h, t \in \mathcal{E}$ denote the head and tail entities, and $r \in \mathcal{R}$ denotes the relation. Finally, we encode each chunk with $E_\theta : \text{text} \to \mathbb{R}^d$ to obtain its embedding $\mathbf{e}_c$. Formally,

$$\mathcal{G} = \{(h, r, t, c, \mathbf{e}_c) \mid (h, r, t) = T_\phi(c), \ c \subset \mathcal{D}, \ \mathbf{e}_c = E_\theta(c)\}. \tag{1}$$

Entities correspond to nodes and relations to edges. Each edge is grounded by its supporting chunk $c$ and its embedding $\mathbf{e}_c$, preserving discrete structure together with contextual semantics and enabling retain/forget control at the level of entities and relations.

**Knowledge Graph-Guided Retrieval.** To enable in-context unlearning, inspired by prior KG-RAG frameworks (Zhu et al., 2025), we design a retrieval process grounded in the knowledge graph $\mathcal{G}$. Given a user query $x$, retrieval proceeds in three stages:

1. **Query encoding.** The query $x$ is parsed by the triple extractor $T_\phi$ into candidate triples, and simultaneously embedded into a dense vector $\mathbf{e}_x = E_\theta(x)$.

2. **Graph matching.** The extracted triples are matched to candidate triples in $\mathcal{G}$ via head/tail text matching, with multi-hop expansion to cover related subgraphs.

3. **Semantic ranking.** Each candidate chunk $c_i$ is encoded to the embedding $\mathbf{e}_{c_i}$. We compute cosine similarity between $\mathbf{e}_x$ and $\mathbf{e}_{c_i}$, *i.e.*, $\text{sim}(x, c_i) = \mathbf{e}_x \cdot \mathbf{e}_{c_i} / (\|\mathbf{e}_x\| \|\mathbf{e}_{c_i}\|)$, and select the top-$k$ chunks as retrieval results.

The retrieved chunks serve as structured evidence for conditioning generation. In the subsequent contrastive decoding stage, these chunks allow the model to selectively suppress information aligned with the forget set while retaining content aligned with the retain set.

## 4 Methodology

We propose *Contrastive Residual Embedding for Decoding (**CRED**)*, an inference-time framework for machine unlearning that requires no parameter updates and is compatible with frozen LLMs. The core idea is to retrieve evidence from separate **retain** and **forget** knowledge graphs, contrast the model behaviors under the two contexts, and inject a residual signal that suppresses the influence of *forget* knowledge while preserving the utility of *retain* knowledge.

The methodology has four parts. First, we construct knowledge conditioned prompts that anchor retain and forget distributions (Section 4.1). Next, we define a token level contrastive residual and describe how it shapes the decoding trajectory (Section 4.2). We then propose a layer-wise fusion mechanism to stabilize residual injection across transformer layers (Section 4.3). Finally, we analyze computational cost, deployment considerations, and limitations, with potential extensions (Section A). Together these components enable practical machine unlearning without retraining.

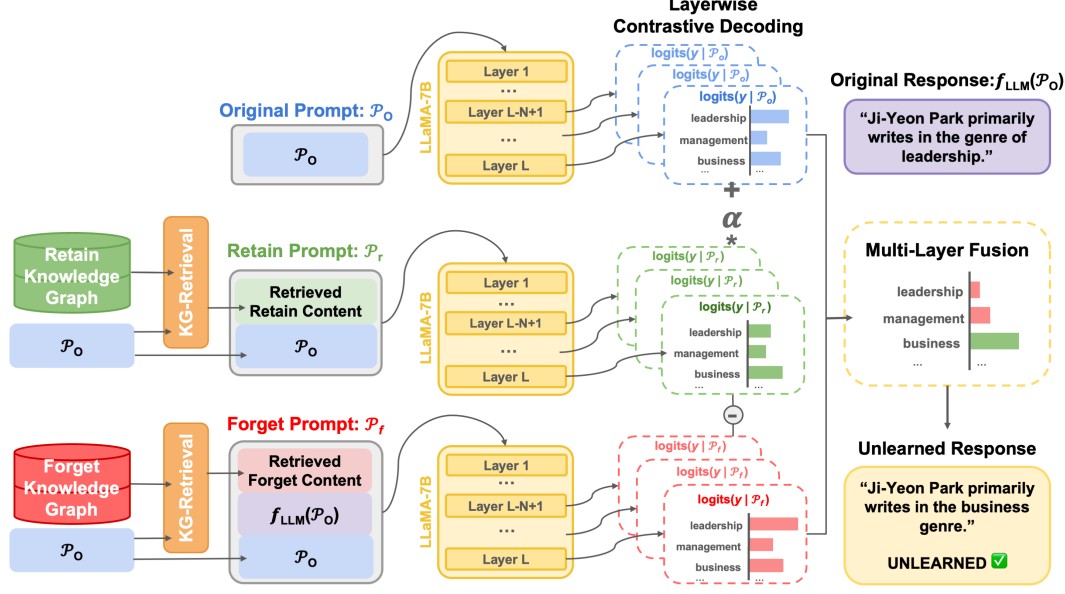

Figure 2: **Overview of CRED**. Given a query, a prompt filter triggers intervention. CRED retrieves evidence from retain and forget knowledge graphs, generates logits conditioned on both, and applies contrastive fusion with the original logits. This suppresses undesired knowledge while preserving retained content, yielding fluent responses without retraining. For details of the prompt-filtering experiments, please refer to the Appendix Section C.

## 4.1 KNOWLEDGE CONDITIONED PROMPT CONSTRUCTION

Let $x$ denote a user query. Given two disjoint knowledge graphs, *i.e.*, a retain graph $\mathcal{G}_r$ and a forget graph $\mathcal{G}_f$, we retrieve topically relevant textual snippets using the *knowledge graph construction* and *knowledge graph-guided retrieval* introduced in Section 3. The relevant textual snippet is expressed by $\mathcal{R}(T_\phi(x), \mathcal{G}, \ell)$ with hop depth $d$. The retrieval procedure is identical for both graphs, ensuring that any differences in the retrieved evidence arise solely from the underlying knowledge split.

To enable precise in-context unlearning, we construct three types of prompts: the original prompt $\mathcal{P}_o$, the retain-conditioned prompt $\mathcal{P}_r$, and the forget-conditioned prompt $\mathcal{P}_f$. $\mathcal{P}_o$ combines a standard system instruction $\mathcal{P}_{SYS}$ with the query $x$, while $\mathcal{P}_r$ and $\mathcal{P}_f$ prepend a retrieval-based instruction $\mathcal{P}_{RAG}$ and include evidence from $\mathcal{G}_r$ and $\mathcal{G}_f$, respectively.

Formally, the prompts are constructed as

$$\begin{aligned}
\mathcal{P}_o &= \mathcal{P}_{SYS} \oplus x, \\
\mathcal{P}_r &= \mathcal{P}_{RAG} \oplus \mathcal{R}(T_\phi(x), \mathcal{G}_r, d) \oplus x, \\
\mathcal{P}_f &= \mathcal{P}_{RAG} \oplus \mathcal{R}(T_\phi(x), \mathcal{G}_f, d) \oplus f_{LLM}(x) \oplus x,
\end{aligned} \quad (2)$$

where $\oplus$ denotes concatenation and $f_{LLM}(x)$ is the frozen LLM's output for $x$. Using only $\mathcal{P}_r$ and $\mathcal{P}_f$ may destabilize generation, so $\mathcal{P}_o$ serves as an anchor to preserve fluency and ensure contrastive adjustments capture semantic differences between retain and forget contexts. It is worth noting that a prompt classifier is first leveraged to determine whether $x$ is associated with the forget set; only queries flagged as forget-related proceed to the CRED decoding stage.

## 4.2 CONTEXT AWARE CONTRASTIVE DECODING

**Token Level Residual.** Let $\mathbf{z}(y|\mathcal{P}) \in \mathbb{R}^{|\mathcal{V}|}$ denote the pre-softmax logits vector for the next token $y$, obtained by applying the language-model head to the decoder state (final layer by default), where $\mathcal{V}$ is the output token set. We first compute

$$\mathbf{z}_o = \mathbf{z}(y|\mathcal{P}_o), \qquad \mathbf{z}_r = \mathbf{z}(y|\mathcal{P}_r), \qquad \mathbf{z}_f = \mathbf{z}(y|\mathcal{P}_f). \qquad (3)$$

We then define the contrastive residual as

$$\Delta \mathbf{z} = \alpha \cdot \left( \mathbf{z}_r - \mathbf{z}_f \right), \tag{4}$$

where $\alpha > 0$ is a scaling factor tuned on a held-out set. The adjusted logits are obtained by

$$\mathbf{z}_{CRED} = \mathbf{z}_o + \Delta \mathbf{z}. \tag{5}$$

Finally, we apply softmax to $\mathbf{z}_{CRED}$ and sample the next token according to the usual decoding scheme (top-$k$, nucleus sampling, or greedy).

It is worth noting that a naive decoding-time defense zeroes logits for tokens that surface in the forget evidence, which fails under paraphrase or sentence-level effects. In contrast, Eq. 5 adjusts the entire logit vector along a semantic direction informed by both retain and forget contexts, suppressing latent dimensions tied to the forget signal while preserving fluency.

## 4.3 LAYER WISE FUSION FOR STABILITY

Decoder layers form a hierarchy: lower layers capture local syntax and higher layers encode global semantics. Applying the residual only at the final layer can hurt grammar if $\alpha$ is large, or leave semantic traces if $\alpha$ is small. Therefore, we distribute the residual across the last $N$ layers. Specifically, let $L$ be the total number of decoder layers and $\mathcal{S}$ define the index set of fused layers, *i.e.*, $\mathcal{S} = \{L - N + 1, \cdots, L\}$. For each $\ell \in \mathcal{S}$, let $\mathbf{z}^{(\ell)}(y|\mathcal{P}) \in \mathbb{R}^{|\mathcal{V}|}$ denote the layer-$\ell$ logits obtained by applying the same LM head to the layer-$\ell$ hidden state for the output token $y$. We form a per-layer adjusted logit vector and then combine:

$$\mathbf{z}_{\text{CRED}}^{(\ell)}(y \mid \mathcal{P}) = \mathbf{z}_o^{(\ell)}(y \mid \mathcal{P}_o) + \alpha \cdot \left[ \mathbf{z}_r^{(\ell)}(y \mid \mathcal{P}_r) - \mathbf{z}_f^{(\ell)}(y \mid \mathcal{P}_f) \right] \tag{6}$$

then form the final logits by a weighted sum

$$\mathbf{z}_{\text{CRED}}(y \mid \mathcal{P}) = \sum_{\ell=L-N+1}^{N} w_\ell \cdot \mathbf{z}_{\text{CRED}}^{(\ell)}(y \mid \mathcal{P}), \quad w_\ell = \frac{\ell - (L - N)}{\sum_{j \in \mathcal{S}} \left( j - (L - N) \right)}. \tag{7}$$

## 5 EXPERIMENT

In this section, we evaluate the proposed **CRED** method against existing baseline approaches on two widely-used LLM unlearning benchmarks. Specifically, we consider **entity unlearning** using the TOFU dataset (Maini et al., 2024), and assess **general unlearning capabilities** through the MUSE-News benchmark (Shi et al., 2025). We further conduct ablation studies to analyze the effectiveness and sensitivity of the contrastive residual decoding mechanism and its key design choices, along with a detailed hyperparameter analysis.

Here, we compare CRED with strong unlearning baselines that span parameter updating and parameter free regimes, grouped as follows.

- **Gradient based methods** update parameters to move the model away from the forget distribution while keeping retain utility. (1) *GA* (Jang et al., 2022; Yao et al., 2024c): anti-train on the forget set to increase its loss to suppress memorization, (2) *GD* (Liu et al., 2022): apply the retain minus forget gradient to erase forget signals while preserving retain behavior, (3) *KL* (Maini et al., 2024): minimize KL divergence on forget inputs toward a neutral reference to flatten logits, (4) *LLMU* (Yao et al., 2024c): finetune with an unlearning objective that weakens forget-set representations while regularizing on retain data to preserve knowledge.

- **Preference based methods** treat unlearning as preference optimization on paired data where retain responses are preferred to forget responses. (1) *PO* (Maini et al., 2024): a generic pairwise preference objective that favors retain responses over forget responses, (2) *DPO* (Rafailov et al., 2023): direct preference optimization that favors retain over forget

Table 1: CRED's performance on TOFU dataset 1%, 5% and 10% using Llama2-7B. FQ, MU, R-RL, and F-RL represent forget quality, model utility, ROUGE-L on the retain dataset, and ROUGE-L on the forget dataset, respectively. We include the original LLM and retain LLM for reference. Baselines marked with an asterisk (*) are taken directly from (Deng et al., 2025). Best/second-best within the final two groups are **bolded**/underlined.

| Method | TOFU 1% | | | | TOFU 5% | | | | TOFU 10% | | | |
|---|---|---|---|---|---|---|---|---|---|---|---|---|
| | FQ (↑) | MU (↑) | F-RL (↓) | R-RL (↑) | FQ (↑) | MU (↑) | F-RL (↓) | R-RL (↑) | FQ (↑) | MU (↑) | F-RL (↓) | R-RL (↑) |
| Original LLM | 1.20e-03 | 0.63 | 0.95 | 0.98 | 5.87e-14 | 0.63 | 0.96 | 0.98 | 4.35e-25 | 0.63 | 0.98 | 0.98 |
| Retain LLM | 1.0 | 0.63 | 0.41 | 0.98 | 1.0 | 0.63 | 0.40 | 0.98 | 1.0 | 0.61 | 0.40 | 0.98 |
| GradAscent* | 6.80e-3 | 0.60 | 0.48 | 0.92 | 8.06e-07 | 0.0 | 3.80e-03 | 3.10e-03 | 5.19e-11 | 0.0 | 0.02 | 0.01 |
| GradDiff* | 6.80e-3 | 0.60 | 0.48 | 0.92 | 2.38e-06 | 0.0 | 4.50e-03 | 4.00e-03 | 7.41e-13 | 0.0 | 0.01 | 0.02 |
| KL* | 3.00e-3 | 0.60 | 0.49 | 0.92 | 4.87e-10 | 0.46 | 1.60e-03 | 0.58 | 4.22e-21 | 0.0 | 0.0 | 0.0 |
| Mismatch* | 6.80e-3 | 0.60 | 0.49 | 0.92 | 2.38e-06 | 0.0 | 4.50e-03 | 4.00e-03 | 7.41e-13 | 0.0 | 0.01 | 0.02 |
| LLMU* | 3.00e-3 | 0.60 | 0.49 | 0.92 | 2.96e-05 | 0.0 | 6.20e-03 | 7.10e-03 | 5.33e-19 | 0.0 | 1.00e-04 | 0.0 |
| PO* | 3.00e-03 | 0.63 | 0.18 | 0.92 | 1.39e-06 | 0.0 | 3.50e-03 | 3.20e-03 | 1.85e-15 | 0.55 | 0.07 | 0.77 |
| DPO* | 0.01 | 0.63 | 0.26 | 0.91 | 1.12e-05 | 0.0 | 0.02 | 0.02 | 2.17e-06 | 0.0 | 0.01 | 0.01 |
| NPO* | 3.00e-03 | 0.60 | 0.51 | 0.93 | 0.18 | 0.30 | 0.33 | 0.40 | 7.3e-03 | 0.05 | 0.17 | 0.20 |
| FLAT (Pearson)* | 0.05 | 0.61 | 0.45 | 0.94 | 4.36e-23 | 0.15 | 0.02 | 0.15 | 5.69e-41 | 0.0 | 0.0 | 0.0 |
| ICUL* | 5.00e-04 | 0.62 | 0.48 | **0.98** | 3.08e-12 | **0.63** | 0.54 | **0.98** | 1.06e-16 | **0.63** | 0.53 | **0.98** |
| Output Filtering* | 2.00e-04 | 0.62 | **0.0** | **0.98** | 5.62e-17 | **0.63** | **6.00-e04** | **0.98** | 1.43e-22 | **0.63** | **1.00e-03** | **0.98** |
| Prompt* | 5.00e-04 | 0.62 | 0.59 | **0.98** | 1.11e-14 | **0.63** | 0.49 | **0.98** | 2.51e-18 | **0.63** | 0.47 | **0.98** |
| GUARD* | 0.17 | 0.62 | 0.39 | **0.98** | 1.83e-05 | **0.63** | 0.40 | **0.98** | 5.73e-07 | **0.63** | 0.40 | **0.98** |
| CRED (ours) | **0.41** | **0.63** | 0.37 | **0.98** | **0.09** | **0.63** | 0.35 | **0.98** | **0.01** | **0.63** | 0.35 | **0.98** |

responses without RL fine-tuning, (3) *NPO* (Zhang et al., 2024a): optimize with negative preferences by explicitly penalizing forget-aligned completions, (4) *FLAT* (Wang et al., 2025c): a correlation-based preference objective that penalizes responses aligned with forget data while reinforcing retain-consistent ones.

- **Training free methods** leave parameters fixed and intervene at prompt or decoding time. (1) *ICUL* (Pawelczyk et al., 2024): in-context unlearning via steering instructions and counter-examples, (2) *Output Filtering* (Thaker et al., 2024): rule-based veto of banned tokens or phrases during generation, (3) *Prompt based strategies* (Deng et al., 2025): negative prompts or constraints that request avoidance of forget content, (4) *GUARD* (Deng et al., 2025): decoding time penalties on forget evidence while preserving fluency.

## 5.1 Entity Unlearning on TOFU

**Experiment Setup.** The TOFU benchmark (Maini et al., 2024) is a synthetic QA dataset designed to evaluate *entity-level unlearning*, which consists of 200 GPT-4-generated fictional author profiles, each with 20 QA pairs (totaling 4,000 pairs). Forget sets are defined at granularities of 1%, 5%, and 10%, corresponding to 2, 10, and 20 authors, respectively. Experiments are performed on the Llama-2-7B-Chat model finetuned on TOFU from Open-Unlearning (Dorna et al., 2025). In our experiments, we set the fusion coefficient $\alpha$ to 0.5 and aggregate representations from the final two layers ($N = 2$), aiming to enhance robustness during unlearning evaluation.

**Evaluation Metrics.** Following prior work (Wang et al., 2025c; Deng et al., 2025), we use four metrics. **Forget Quality (FQ)** is the KS test p-value between outputs of the unlearned and Retain LLMs on the forget set (higher indicates behavior closer to Retain LLM). **Model Utility (MU)** measures performance on the retain set, world facts, and real authors. **ROUGE-L** (Lin, 2004) is reported on the forget set (**F-RL**) and retain set (**R-RL**) to evaluate removal and preservation quality.

**Main Results on TOFU.** Table 1 demonstrates that CRED achieves state-of-the-art performance across all TOFU splits. Across all metrics, CRED consistently outperforms existing in-context unlearning methods, with particularly strong gains in forgetting quality, indicating that it effectively removes undesired knowledge while maintaining overall model behavior. The improvement is stable across different deletion granularities, underscoring the robustness of our approach.

CRED achieves superior Forget Quality (FQ), with forget-set behavior most closely aligned to the Retain LLM. Compared with all baselines, including alternative in-context methods with similar

Table 2: CRED's MUSE performance across four criteria, with lower values preferred on $D_f$ and higher values on $D_r$. For PrivLeak, values closer to 0 indicate better privacy. Results are shown in ✔ when the criterion is satisfied and in ✗ otherwise. Baselines with an asterisk (*) are from (Deng et al., 2025). All VerbMem and KnowMem values are reported as percentages (%).

| Method | VerbMem on $D_f$ ($\downarrow$) | KnowMem on $D_f$ ($\downarrow$) | KnowMem on $D_r$ ($\uparrow$) | PrivLeak ($\to 0$) |
|---|---|---|---|---|
| Original LLM | 57.9 | 64.4 | 55.5 | -99.8 |
| Retained LLM | 20.2 | 32.7 | 56.0 | 0.0 |
| GradAscent* | 0.0 (✔) | 0.0 (✔) | 0.0 (✗) | 17.0 |
| GradDiff* | 4.9 (✔) | 27.5 (✔) | 6.7 (✗) | 109.4 |
| KL* | 27.4 (✗) | 50.2 (✗) | 44.8 (✗) | -96.1 |
| NPO* | 0.0 (✔) | 0.0 (✔) | 0.0 (✗) | 15.0 |
| NPO-RT* | 1.2 (✔) | 54.6 (✗) | 40.5 (✗) | 105.8 |
| FLAT (Pearson)* | 1.6 (✔) | 0.0 (✔) | 0.2 (✗) | 26.8 |
| ICUL* | 10.7 (✔) | 19.7 (✔) | 55.2 (✔) | -99.8 |
| Output Filtering* | **1.1** (✔) | 0.3 (✔) | 55.2 (✔) | -99.8 |
| Prompt* | 15.4 (✔) | 47.9 (✗) | 55.2 (✔) | -99.6 |
| GUARD* | 4.3 (✔) | 4.9 (✔) | 55.2 (✔) | 109.6 |
| **CRED (ours)** | 18.0 (✔) | **4.0** (✔) | **55.5** (✔) | **36.9** |

utility and retention, CRED demonstrates substantially stronger forgetting performance, underscoring its practical effectiveness.

## 5.2 MUSE-NEWS RESULTS

**Experiment Setup.** We further evaluate CRED on MUSE-News, a realistic benchmark using BBC news articles post-August 2023 with disjoint forget, retain, and holdout sets. Experiments use the pretrained **Llama2-7B** model from MUSE (Shi et al., 2025). We adopt a fusion strategy with coefficient $\alpha = 4.0$, combining the top two hidden layers to improve evaluation stability (see Appendix D.1 for analysis).

**Evaluation Metrics.** We follow the MUSE protocol (Shi et al., 2025). **VerbMem**(ROUGE-L F1) measures verbatim memorization via a text-completion task on the forget set, testing whether the model reproduces removed passages. **KnowMem**(ROUGE-L F1) evaluates factual retention through QA-style assessment on both the forget and retain sets. **PrivLeak** quantifies privacy risk via membership inference attacks (MIA) (Murakonda et al., 2021; Ye et al., 2022), which attempt to infer whether a sample from the forget set was part of training by exploiting loss statistics. Following prior work, we measure privacy leakage as the difference in AUC-ROC between the unlearned and fully retrained models, where lower values indicate reduced residual training signals.

**Discussion on MUSE-News.** As shown in Table 2, CRED achieves state-of-the-art forgetting and privacy on MUSE-News. It obtains the lowest **KnowMem** on the forget set, showing effective removal of forgotten knowledge while maintaining overall model performance. For privacy, CRED achieves the strongest **PrivLeak** among training-free methods, indicating that its decoding-time mechanism substantially suppresses residual membership signals without retraining or hard filtering.

For **VerbMem**, filtering and prompt-based methods reduce scores by blocking verbatim content, and GUARD applies semantic blocking, but all risk unnatural outputs. In contrast, CRED implicitly attenuates the *conceptual* knowledge of the forget set during decoding, achieving effective forgetting without degrading usability. This balance between forgetting, privacy, and natural response generation underscores its advantage over prior in-context unlearning approaches.

## 5.3 QUANTIZATION CONSISTENCY AND STABILITY ANALYSIS

We evaluate robustness to quantization on the **MUSE-News** benchmark because practical deployments often run LLMs in reduced precision to lower latency. Previous work (Zhang et al., 2025)

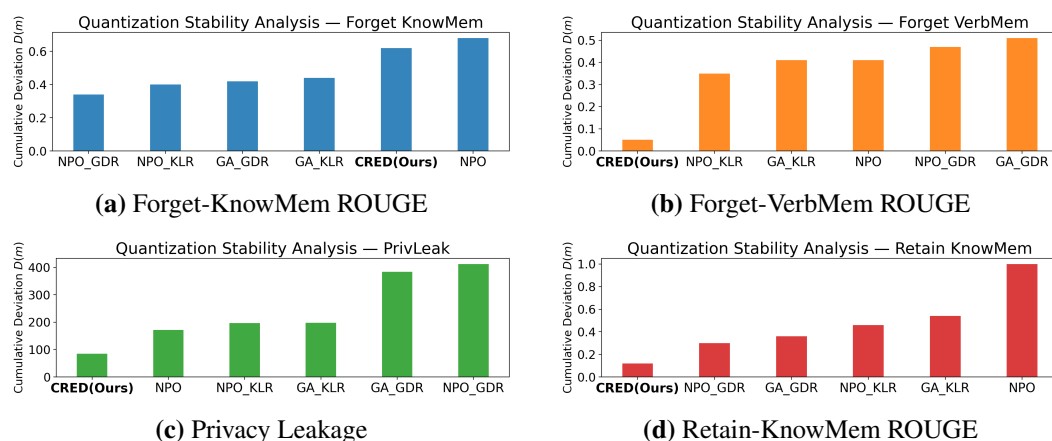

**(a)** Forget-KnowMem ROUGE

**(b)** Forget-VerbMem ROUGE

**(c)** Privacy Leakage

**(d)** Retain-KnowMem ROUGE

Figure 3: Quantization deviation across four metrics in the order (a)–(d). Lower values indicate greater consistency with the retain baseline and stronger stability under reduced precision.

shows that quantization can distort hidden states and logits. As such, unlearning methods that rely on small residual adjustments may therefore be brittle. We test each method at full precision, 8-bit, and 4-bit inference, and report a *quantization deviation* that reflects both self consistency and performance relative to a retain-only baseline (no unlearning) under the same precision:

$$\Delta_z^{\mathrm{R}}(m) = \left| s_{\mathrm{retain},z} - s_{m,z} \right|, \qquad \Delta_z^{\mathrm{Q}}(m) = \left| s_{m,\mathrm{base}} - s_{m,z} \right|, \tag{8}$$

and the cumulative deviation as

$$D(m) = \sum_{z \in \{8\text{-}bit,\, 4\text{-}bit\}} \left( \Delta_z^{\mathrm{R}}(m) + \Delta_z^{\mathrm{Q}}(m) \right). \tag{9}$$

Here, $\Delta_z^{\mathrm{R}}(m)$ quantifies deviation from the retain baseline at precision $z$, indicating whether forgetting remains comparable to the retain model under quantization. In contrast, $\Delta_z^{\mathrm{Q}}(m)$ measures deviation from the method's own unquantized model, reflecting robustness to precision changes. Together, they capture cross-method consistency and intra-method stability under quantization.

**Quantization Stability.** Figure 3 shows that CRED achieves the lowest deviation under both 8-bit and 4-bit quantization, whereas other methods fluctuate greatly, especially on privacy leakage. This robustness stems from its training-free, parameter-agnostic design, ensuring stable forgetting across metrics in resource-constrained settings. The larger deviation in subplot (a) is due to CRED's much lower KnowMem on $D_f$ compared to Retain (Table 10), yet its forgetting performance remains consistent across precisions.

## 5.4 ABLATION STUDY

**Normalization.** For completeness, we evaluate two normalization operators applied to each logit vector (Eq. 3) before contrastive fusion:

$$\tilde{\mathbf{z}}_{\mathrm{o}} = \mathrm{Norm}(\mathbf{z}_{\mathrm{o}}), \qquad \tilde{\mathbf{z}}_{\mathrm{r}} = \mathrm{Norm}(\mathbf{z}_{\mathrm{r}}), \qquad \tilde{\mathbf{z}}_{\mathrm{f}} = \mathrm{Norm}(\mathbf{z}_{\mathrm{f}}) \tag{10}$$

where $\mathrm{Norm} \in \{\mathrm{Softmax},\ \mathrm{Min}-\mathrm{Max}\}$ denotes the normalization method applied independently to the original, retain, and forget logits. Table 3 (A) shows a clear and consistent trend: applying either softmax or min–max normalization severely weakens forgetting. The unnormalized variant achieves the strongest FQ across all TOFU splits, whereas both normalization methods drive FQ to near-zero levels. This confirms that retaining the raw logit scale is essential for preserving effective contrastive signals, and unnormalized logits remain the most reliable choice.

**Retain-Guided Contrast for Stable Unlearning.** As shown in Table 3 (B), the retain branch plays a decisive role in ensuring the stability of the unlearning process. Without retain guidance, the

Table 3: Ablation study across three dimensions: (A) logit normalization, (B) retain–forget contrastive structure, and (C) fixed vs. adaptive $\alpha$. Evaluated on TOFU (1%, 5%, 10%) with `Llama-2-7B-Chat`; higher FQ and lower F-RL are better.

| Method | TOFU 1% | | TOFU 5% | | TOFU 10% | |
|---|---|---|---|---|---|---|
| | FQ ($\uparrow$) | F-RL ($\downarrow$) | FQ ($\uparrow$) | F-RL ($\downarrow$) | FQ ($\uparrow$) | F-RL ($\downarrow$) |
| *(A) Logit Normalization Strategies* | | | | | | |
| **w/o norm** | **0.41** | **0.37** | **0.09** | **0.35** | **0.01** | **0.35** |
| softmax | 1.9e-6 | 0.68 | 4.9e-25 | 0.63 | 8.2e-46 | 0.65 |
| min-max | 5.0e-7 | 0.47 | 8.2e-34 | 0.44 | 4.1e-69 | 0.45 |
| *(B) Retain Knowledge for CRED* | | | | | | |
| **w/ retain** | **0.41** | **0.37** | **0.09** | **0.35** | **0.01** | **0.35** |
| w/o retain | 0.03 | 0.38 | 1.4e-11 | **0.35** | 4.3e-23 | **0.35** |
| *(C) $\alpha$ Strategy* | | | | | | |
| **fixed** $\alpha$ | **0.41** | **0.37** | **0.09** | **0.35** | **0.01** | **0.35** |
| adaptive $\alpha$ | 0.01 | 0.39 | 2.4e-10 | 0.37 | 1.2e-17 | 0.38 |

model relies solely on repulsion from the forget distribution, which results in an ill-posed optimization direction and severe semantic degradation, leading to a dramatic collapse in FQ. In contrast, introducing a retain–forget contrastive structure imposes a bidirectional constraint: it not only enforces effective suppression of the target knowledge but also anchors decoding toward semantically appropriate and fluent regions of the distribution. This retain-guided contrast serves as a core stabilizing mechanism in CRED, enabling the model to achieve strong forgetting performance without sacrificing generation quality.

**Adaptive $\alpha$ computation.** In the adaptive variant, the fusion coefficient $\alpha$ is computed from the cosine similarity between the logits of the original prompt and the retrieved forget prompt:

$$\alpha = \text{cosine\_sim}\Big( \mathbf{z}_o^{(\ell)}(y \mid \mathcal{P}_o),\ \mathbf{z}_f^{(\ell)}(y \mid \mathcal{P}_f) \Big), \tag{11}$$

where $\mathbf{z}_o^{(\ell)}(y \mid \mathcal{P}_o)$ denotes the layer-$\ell$ logits produced when the model processes the original prompt $\mathcal{P}_o$, and $\mathbf{z}_f^{(\ell)}(y \mid \mathcal{P}_f)$ denotes the corresponding logits under the retrieved forget prompt $\mathcal{P}_f$. For completeness, the normalization term $Z^*$ follows the contrastive logits formulation introduced in the main text; please refer to Section 4.2. This adaptive setting provides a dynamic scaling factor that modulates the residual strength based on how closely the two logit vectors align at each decoding step.

# 6 CONCLUSION

We introduced **CRED**, an inference-time unlearning framework that integrates knowledge graphs with contrastive residual decoding to enable precise, controllable forgetting without retraining or parameter edits. By operating entirely during decoding, CRED suppresses targeted concepts while preserving unaffected knowledge, making it lightweight and deployment-ready. Experiments on TOFU and MUSE-News demonstrate strong forgetting and privacy performance compared to both training-based and inference-time baselines—achieving more than threefold higher forget quality on TOFU with maintained utility, and competitive results on MUSE. We further evaluated robustness under 8-bit and 4-bit quantization with a simple deviation measure, showing consistently low deviation across metrics and indicating that unlearning and utility remain stable under compression. Additional ablations confirm transferability across model architectures and sizes, and the KG-centric design allows efficient updates to the forget/retain sets without parameter changes. Together, these results position CRED as a flexible and scalable approach to precise, compliant, and deployment-ready unlearning. Future work will explore automatic query detection, extensions to multilingual and multimodal domains, adaptive scaling of the fusion coefficient $\alpha$, and quantization-aware decoding strategies tailored for efficient real-world use.

## REPRODUCIBILITY STATEMENT

To facilitate reproducibility, we provide our implementation in an anonymous code repository: `https://anonymous.4open.science/r/CRED-C969`. The repository includes the core code and configuration files that specify all hyperparameters used in our experiments. Preprocessed datasets and additional resources will be released upon acceptance of the paper. Detailed descriptions of the experimental settings and procedures are provided in the main text and appendix.

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

Table 4: Relative throughput comparison between optimized CRED and baseline inference across batch sizes (up to 24), measured with KV-cache enabled and generating 32 tokens. The optimized CRED incurs an overhead of about $\sim$1.3–2.4$\times$, while maintaining stable scaling across batch sizes.

| Batch | Input tokens/Batch size | CRED (tokens/sec) | Baseline (tokens/sec) |
|---|---|---|---|
| 4 | 615.2 | 142.1 | 179.2 |
| 8 | 622.1 | 178.2 | 317.4 |
| 12 | 626.2 | 195.3 | 394.1 |
| 16 | 627.9 | 203.3 | 443.8 |
| 20 | 634.4 | 208.2 | 480.7 |
| 24 | 638.3 | 210.3 | 508.0 |

# A    COMPLEXITY AND DEPLOYMENT CONSIDERATIONS

## A.1    COMPUTATION AND MEMORY OVERHEAD.

CRED requires two additional forward passes per query (retain/forget contexts). On **Llama-2-7B** with a single **NVIDIA H100 80GB** GPU, the optimized implementation incurs an overhead of about **1.3–2.4**$\times$ relative to the baseline inference, while maintaining stable scaling across batch sizes (Table 4). As such, the memory overhead remains modest (less than 96MB per query with mixed precision and vocabulary of 4k), easily supported by modern hardware. As a fully prompt-driven approach, CRED also enables rapid adaptation to new removal requests by simply updating the forget graph $\mathcal{G}_f$, without the need for retraining or hyperparameter tuning.

## A.2    LIMITATIONS AND FUTURE EXTENSIONS.

When retain and forget concepts exhibit significant overlap, the fixed scaling coefficient $\alpha$ in Eq. (4) may inadvertently suppress useful information along with the targeted knowledge. A promising future direction is to make $\alpha$ adaptive to each query $q$, namely, $\alpha(q)$, so that the strength of the contrastive residual can be dynamically modulated based on query sensitivity or semantic similarity. Alternatively, projecting the residual onto concept-specific subspaces could further improve selectivity and robustness. In summary, CRED combines structured retrieval and contrastive decoding to achieve practical, parameter-free machine unlearning. The method preserves language quality, effectively removes latent concept influence, and integrates seamlessly with existing deployment pipelines. The following sections provide quantitative and qualitative evidence for its effectiveness.

# B    PROMPT DESIGN DETAILS

**Retrieval-based System Prompt.**    The retrieval-based system prompt $\mathcal{P}_{RAG}$ restricts the model to retrieved passages only, preventing reliance on inherent knowledge or speculation and ensuring that differences between $\mathcal{P}_r$ and $\mathcal{P}_f$ derive solely from their respective knowledge graphs. The exact system prompt used in our retrieval-augmented generation experiments is shown below:

> You are a RAG-based question answering assistant.
>
> Please strictly and only refer to the source content below to answer the question concisely. Do not use any prior knowledge, commonsense reasoning, or subjective judgment. If the source content does not contain information relevant to the question, reply: No relevant information provided.
>
> Answering Guidelines:
>
> - Base your answer exclusively on the source content. Do not use the model's own knowledge or add any assumptions.
> - Do not express opinions, give explanations, perform analysis, or elaborate.
> - Only include information directly relevant to the question; avoid unnecessary details.
> - Ensure factual accuracy with no fabrication or errors.
> - If no relevant information is found in the source content, reply: No relevant information provided.

Table 5: Token Counts Used for Training and Testing the Prompt Classifier

| Dataset | $\mathbf{D_P^{Train}}$ | $\mathbf{D_N^{Train}}$ | $\mathbf{D_P^{Test}}$ | $\mathbf{D_N^{Test}}$ |
|---|---|---|---|---|
| TOFU (1%) | 1K | 72K | 1K | 13K |
| TOFU (5%) | 4K | 69K | 4K | 13K |
| TOFU (10%) | 8K | 65K | 8K | 13K |
| MUSE-News (knowmem) | 16K | 16K | 2K | 2K |
| MUSE-News (verbmem) | 2953K | 1446K | 90K | – |

Table 6: Error Rates of the Prompt Classifier on Training and Testing Sets

| Dataset | FPR @ $D^{Train}$ | FNR @ $D^{Train}$ | FPR @ $D^{Test}$ | FNR @ $D^{Test}$ |
|---|---|---|---|---|
| TOFU (1%) | 0.0 | 0.0 | 0.0 | 0.0 |
| TOFU (5%) | 0.0 | 0.0 | 0.0 | 0.0 |
| TOFU (10%) | 0.0 | 0.0 | 0.0 | 0.0 |
| MUSE-News (knowmem) | 0.0 | 0.0 | 0.0 | 0.0 |
| MUSE-News (verbmem) | 0.029 | 0.011 | – | 0.017 |

## C  PROMPT CLASSIFIER

Following prior work on classifier-guided unlearning (Deng et al., 2025; Liu et al., 2024), we introduce a lightweight prompt classifier that distinguishes forget-related queries from regular user prompts. The classifier provides an auxiliary signal indicating whether a prompt should trigger unlearning-specific decoding behavior.

We employ a RoBERTa-base encoder and fine-tune it on dataset-specific positive (forget) and negative (retain) prompts. Separate classifiers are trained for each unlearning dataset, and they are evaluated on both in-distribution and perturbed test prompts. The subsequent subsections present detailed training configurations and experimental results for TOFU and MUSE-News.

### C.1  PROMPT CLASSIFIER DATASET AND EXPERIMENTS ON TOFU

For the TOFU dataset (Maini et al., 2024), we follow the original forget–retain splits at 1%, 5%, and 10% to construct the training data for the prompt classifier. The forget subsets serve as positive samples, while the corresponding retain subsets serve as negative samples.

To evaluate false positives and generalization beyond the training distribution, we use the standard TOFU evaluation splits, including `real_authors`, `world_facts`, and their perturbed variants. Perturbed forget prompts are also included to assess robustness under distribution shift. The token counts and classification error rates are reported in Tables 5 and 6.

### C.2  PROMPT CLASSIFIER DATASET AND EXPERIMENTS ON MUSE-NEWS

The MUSE-News benchmark (Shi et al., 2025) contains two tasks, *knowmem* and *verbmem*, for which we train separate prompt classifiers. For the *knowmem* setting, we follow the data construction procedure described in GUARD (Deng et al., 2025) and apply paraphrasing-based augmentation to the QA questions using an open-source LLM. Forget samples are treated as positive and retain samples as negative. As shown in Tables 5 and 6, the classifier yields zero error in this setting.

For the *verbmem* setting, we follow the training data provided by MUSE-News and adopt the sentence-level construction described in (Liu et al., 2024). Each document is split into sentences using spaCy's sentencizer (Honnibal et al., 2020), retaining only those longer than 20 characters. The resulting sentence-level samples are used directly for training and evaluation. Although the classifier exhibits non-zero errors at the sentence level, applying group voting over the original (pre-segmentation) text blocks results in perfect classification performance.

Table 7: Hyperparameter study of CRED layer fusion, examining how injecting residual signals into the last $N$ decoder layers.

| Layers | TOFU 1% | | TOFU 5% | | TOFU 10% | |
|---|---|---|---|---|---|---|
| | FQ (↑) | F-RL (↓) | FQ (↑) | F-RL (↓) | FQ (↑) | F-RL (↓) |
| Last 1 | 0.40 | 0.45 | 0.02 | 0.43 | 0.002 | 0.44 |
| **Last 2** | **0.41** | **0.37** | **0.09** | **0.35** | **0.010** | **0.35** |
| Last 3 | 0.40 | 0.44 | 0.07 | 0.42 | 0.010 | 0.42 |
| Last 4 | 0.40 | 0.43 | 0.07 | 0.41 | 0.002 | 0.41 |

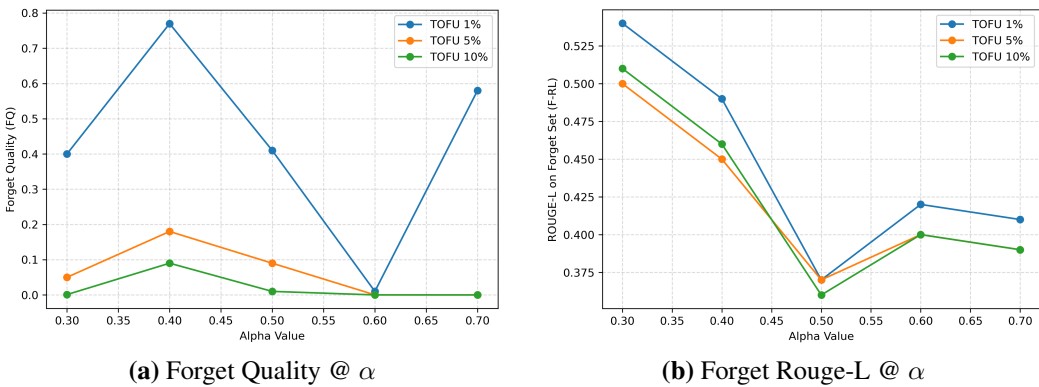

**(a)** Forget Quality @ $\alpha$  **(b)** Forget Rouge-L @ $\alpha$

Figure 4: Effect of $\alpha$ on (a) Forget Quality and (b) Rouge Loss on the Forget Set for TOFU 1%, 5%, and 10% with `Llama-2-7B-Chat`. Higher FQ in (a) and lower F-RL in (b) indicate better forgetting. $\alpha = 0.5$ achieves the best trade-off and is used in the final experiments.

# D  SENSITIVITY ANALYSIS

## D.1  CONTRASTIVE FUSION ANALYSIS

**Fusion Coefficient Analysis on TOFU.**  Figure 4 illustrates the effect of the scaling coefficient $\alpha$ on Forget Quality (FQ) and ROUGE-L on the Forget Set (F-RL) across different removal ratios in TOFU. Increasing $\alpha$ from 0.3 to 0.5 substantially improves forgetting, but further increases yield diminishing or even negative returns, as the residual begins to overpower the model's native generation. Overall, $\alpha = 0.5$ achieves the best trade-off between forgetting and output fluency.

**Layer Fusion Depth Analysis on TOFU**  Table 7 shows that injecting residuals only into the last one layer yields insufficient forgetting, while extending to the last two layers substantially boosts FQ and reduces F-RL. Expanding beyond two layers provides no further gains and even disrupts utility, indicating that Last-2 offers the most stable and effective fusion depth.

**Fusion Coefficient Analysis on MUSE-News.**  On the MUSE-News benchmark, Table 8 shows smaller values of $\alpha$ weaken forgetting performance and increase privacy leakage. As $\alpha$ grows, forgetting improves steadily while privacy leakage stabilizes. We therefore select $\alpha = 4.0$ as the default, as it balances effective forgetting with stable privacy protection.

**Summary.**  Across both TOFU and MUSE-News, the results suggest that contrastive fusion benefits from moderate scaling. Very small $\alpha$ values are insufficient for strong forgetting, while overly large values risk unstable or degenerate outputs. A mid-range $\alpha$ (0.5 for TOFU, 4.0 for MUSE) provides the best balance between forgetting, retention, and privacy.

## D.2  KNOWLEDGE GRAPH EXTRACTION HYPERPARAMETER ANALYSIS

**Effect of Hop Depth.**  As shown in Table 9, increasing the hop depth—i.e., expanding the extracted subgraph to include nodes at larger in/out-degree distances from the query entity—does not

Table 8: Hyperparameter tuning of the scaling coefficient $\alpha$ on the MUSE-News benchmark with `Llama-2-7B`. Lower values on $D_f$ indicate better forgetting, and PrivLeak values closer to 0 indicate stronger privacy. VerbMem and KnowMem are reported as percentages (%).

| Method | VerbMem on $D_f$ ($\downarrow$) | KnowMem on $D_f$ ($\downarrow$) | PrivLeak ($\rightarrow 0$) |
|---|---|---|---|
| Target | 57.9 | 64.4 | -99.8 |
| Retain | 20.2 | 32.7 | 0.0 |
| CRED$_{\alpha=0.5}$ | 40.0 | 30.8 | 36.2 |
| CRED$_{\alpha=1.0}$ | 29.9 | 10.3 | 36.2 |
| CRED$_{\alpha=1.5}$ | 26.7 | 6.3 | 36.2 |
| CRED$_{\alpha=2.0}$ | 26.1 | 8.6 | 36.9 |
| CRED$_{\alpha=2.5}$ | 22.7 | 6.7 | 36.9 |
| CRED$_{\alpha=3.0}$ | 21.1 | 6.9 | 36.9 |
| CRED$_{\alpha=3.5}$ | 20.2 | 4.9 | 36.9 |
| CRED$_{\alpha=4.0}$ | 18.0 | 4.0 | 36.9 |
| CRED$_{\alpha=4.5}$ | 18.0 | 4.1 | 36.3 |
| CRED$_{\alpha=5.0}$ | 16.3 | 3.4 | 36.3 |

Table 9: Comparison of knowledge-graph retrieval hyperparameters on TOFU using Llama-2-7B-Chat. We evaluate hop depth and Top-k node selection under different forgetting ratios.

| Setting | TOFU 1% | | TOFU 5% | | TOFU 10% | |
|---|---|---|---|---|---|---|
| | FQ ($\uparrow$) | F-RL ($\downarrow$) | FQ ($\uparrow$) | F-RL ($\downarrow$) | FQ ($\uparrow$) | F-RL ($\downarrow$) |
| *(Hop = 1)* | | | | | | |
| Top-1 | 0.40 | 0.44 | 0.05 | 0.42 | 0.008 | 0.43 |
| Top-2 | 0.40 | 0.44 | 0.07 | 0.42 | 0.006 | 0.42 |
| **Top-3** | **0.41** | **0.37** | **0.09** | **0.35** | **0.010** | **0.35** |
| *(Hop = 2)* | | | | | | |
| Top-1 | 0.40 | 0.44 | 0.03 | 0.43 | 0.008 | 0.43 |
| Top-2 | 0.40 | 0.45 | 0.03 | 0.42 | 0.008 | 0.43 |
| Top-3 | 0.40 | 0.45 | 0.07 | 0.42 | 0.010 | 0.42 |

improve forgetting performance. Under the same Top-$k$ setting, 1-hop neighborhoods already provide sufficient local structure for identifying forget-related triples, while 2-hop expansion pulls in many peripheral or weakly related nodes. This additional noise reduces the usefulness of the extracted subgraph and leads to no observable gains across all TOFU splits.

**Effect of Top-k.** In the 1-hop setting, a larger Top-$k$ generally improves forgetting, as it provides richer but still relevant context for contrastive fusion. As shown in Table 9, Top-$k = 3$ yields the best overall results across all TOFU splits. Smaller $k$ under-extracts useful attributes, while larger $k$ introduces unnecessary noise without further gains.

**Summary.** A shallow 1-hop neighborhood provides cleaner and more relevant graph structure, while a moderate Top-$k$ ensures sufficient contextual coverage without introducing noise. Therefore, we adopt **1-hop with Top-k=3** as the default extraction setting.

# E   CONSISTENCY AND ROBUSTNESS UNDER QUANTIZATION

Table 10 reports the quantization stability results on MUSE-News. CRED demonstrates highly consistent performance across full precision, 8-bit, and 4-bit inference, with only minimal variation in both forgetting and retention metrics. This stability verifies that our training-free and parameter-agnostic design remains effective under reduced precision, ensuring that unlearning performance is preserved even when the model is compressed. These findings highlight the robustness of CRED and its suitability for deployment in resource-constrained scenarios.

Table 10: Quantization stability results on MUSE-News with `Llama-2-7B`. Values are in percentage except for *PrivLeak*. *Target*, *Retain*, and *CRED* are our implementations. Baselines marked with an asterisk (*) are adopted from (Zhang et al., 2025).

| Method | VerbMem on $D_f$ ($\downarrow$) | KnowMem on $D_f$ ($\downarrow$) | KnowMem on $D_r$ ($\uparrow$) | PrivLeak ($\rightarrow$ 0) |
|---|---|---|---|---|
| Target | 66.0 | 57.0 | -99.8 | 55.0 |
| Target + Quan. (8 bit) | 65.0 | 57.0 | -99.8 | 56.0 |
| Target + Quan. (4 bit) | 55.0 | 49.0 | -99.8 | 48.0 |
| Retain | 32.0 | 20.0 | 0.0 | 56.0 |
| Retain + Quan. (8 bit) | 32.0 | 21.0 | -0.4 | 54.0 |
| Retain + Quan. (4 bit) | 36.0 | 20.0 | -2.1 | 46.0 |
| GA_GDR* | 29.0 | 0.0 | 87.1 | 34.0 |
| GA_GDR* + Quan. (8 bit) | 27.0 | 0.0 | 93.5 | 34.0 |
| GA_GDR* + Quan. (4 bit) | 50.0 | 25.0 | -99.1 | 48.0 |
| GA_KLR* | 27.0 | 14.0 | -91.6 | 23.0 |
| GA_KLR* + Quan. (8 bit) | 29.0 | 15.0 | -91.7 | 25.0 |
| GA_KLR* + Quan. (4 bit) | 51.0 | 34.0 | -99.8 | 46.0 |
| NPO* | 0.0 | 0.0 | 14.5 | 0.0 |
| NPO* + Quan. (8 bit) | 0.0 | 0.0 | 15.0 | 0.0 |
| NPO* + Quan. (4 bit) | 25.0 | 16.0 | -71.6 | 28.0 |
| NPO_GDR* | 46.0 | 30.0 | 107.2 | 39.0 |
| NPO_GDR* + Quan. (8 bit) | 44.0 | 10.0 | 106.3 | 37.0 |
| NPO_GDR* Quan. (4 bit) | 51.0 | 33.0 | -99.8 | 48.0 |
| NPO_KLR* | 37.0 | 17.0 | -94.0 | 33.0 |
| NPO_KLR* + Quan. (8 bit) | 37.0 | 17.0 | -93.7 | 30.0 |
| NPO_KLR* + Quan. (4 bit) | 54.0 | 34.0 | -99.8 | 49.0 |
| CRED | 4.0 | 18.0 | 36.9 | 55.0 |
| CRED + Quan. (8 bit) | 4.0 | 18.0 | 37.8 | 56.0 |
| CRED + Quan. (4 bit) | 3.0 | 18.0 | 40.0 | 48.0 |

# F EXTENDED RESULTS: ACROSS DIFFERENT MODEL SIZES AND BACKBONES ON TOFU BENCHMARK

Table 11 reports large-scale TOFU evaluations across multiple model sizes and backbones. Several consistent patterns emerge.

**Effect of Forget Ratio.** The forget ratio reflects how much domain-specific knowledge the Original LLM has acquired. As the ratio increases, the model's predictions on the forget set become more stable and semantically concentrated, making it increasingly difficult to shift its output distribution toward that of the Retain LLM. This trend is clearly observed across all backbones: for a fixed model, *higher forget ratios consistently lead to lower FQ values*. This indicates that unlearning difficulty scales directly with the amount of knowledge the model has memorized.

**Effect of Model Scale.** Across all model sizes (1B–70B), CRED maintains *Retain-level F-RL*, demonstrating that generation-level suppression of forgotten knowledge is stable and predictable. However, Forget Quality (FQ) exhibits a strong scaling effect: *larger models retain more rigid and stable internal representations*, making their output distributions harder to realign with the Retain LLM. This explains the uniformly low FQ values observed for large Llama models. Notably, Llama-3.3-70B is highly resistant to distributional adjustments and requires a significantly stronger intervention ($\alpha = 2.0$)—in contrast to $\alpha = 0.5$ for smaller models—to achieve measurable forgetting.

**Backbone Differences.** Qwen-3-32B displays trends that closely mirror the Llama family: (1) FQ decreases as model size or forget ratio increases; (2) F-RL remains aligned with the Retain LLM; and (3) larger models remain harder to forget. This demonstrates that CRED generalizes effectively across architectures, and the observed scaling behavior is a fundamental property of large Transformer models rather than a Llama-specific phenomenon.

Table 11: Comparison of forget quality (FQ) and forget ROUGE-L (F-RL) across multiple backbone sizes on the TOFU benchmark under Original, Retain, and CRED

| Backbone | Original LLM | | Retain LLM | | CRED | |
|---|---|---|---|---|---|---|
| | FQ ($\uparrow$) | F-RL ($\downarrow$) | FQ ($\uparrow$) | F-RL ($\downarrow$) | FQ ($\uparrow$) | F-RL ($\downarrow$) |
| **TOFU 1%** | | | | | | |
| Llama-3.2-1B | 0.007 | 0.87 | 1.0 | 0.41 | 0.92 | 0.42 |
| Llama-2-7B | 1.20e-03 | 0.95 | 1.0 | 0.41 | 0.58 | 0.39 |
| Llama-3.1-8B | 0.007 | 0.99 | 1.0 | 0.42 | 0.03 | 0.33 |
| Llama-3.3-70B$_{\alpha=2.0}$ | 0.014 | 0.99 | 1.0 | 0.40 | 5.01e-07 | 0.40 |
| Qwen 3-32B | 0.003 | 0.98 | 1.0 | 0.45 | 0.001 | 0.40 |
| **TOFU 5%** | | | | | | |
| Llama-3.2-1B | 1.42e-12 | 0.83 | 1.0 | 0.38 | 0.63 | 0.42 |
| Llama-2-7B | 5.87e-14 | 0.96 | 1.0 | 0.40 | 0.04 | 0.37 |
| Llama-3.1-8B | 6.54e-13 | 0.99 | 1.0 | 0.39 | 3.6e-09 | 0.34 |
| Llama-3.3-70B$_{\alpha=2.0}$ | 2.44e-10 | 0.99 | 1.0 | 0.40 | 3.86e-28 | 0.39 |
| Qwen 3-32B | 6.57e-12 | 0.93 | 1.0 | 0.44 | 5.87e-14 | 0.41 |
| **TOFU 10%** | | | | | | |
| Llama-3.2-1B | 3.9e-22 | 0.82 | 1.0 | 0.38 | 0.90 | 0.41 |
| Llama-2-7B | 4.35e-25 | 0.98 | 1.0 | 0.40 | 0.005 | 0.36 |
| Llama-3.1-8B | 1.59e-27 | 0.99 | 1.0 | 0.39 | 1.12e-19 | 0.34 |
| Llama-3.3-70B$_{\alpha=2.0}$ | 8.97e-32 | 0.99 | 1.0 | 0.38 | 8.23e-54 | 0.39 |
| Qwen 3-32B | 2.81e-20 | 0.93 | 1.0 | 0.43 | 2.41e-29 | 0.39 |

**Comparison to Baselines.** Across all settings, CRED consistently outperforms the Original LLM by substantially reducing F-RL while maintaining performance close to the Retain LLM. Even in challenging conditions—large models and high forget ratios—CRED provides controlled and predictable forgetting without causing unnecessary over-forgetting or harming retained content.

## G  USE OF LARGE LANGUAGE MODELS

In accordance with the submission requirements, we provide a detailed disclosure of how Large Language Models (LLMs) were used during the development of this work. Their role was limited to auxiliary support; all scientific ideas, experimental designs, and conclusions were conceived, implemented, and validated by the authors.

LLMs contributed in the following ways:

- **Writing aid and stylistic refinement.** LLMs were used to improve clarity, coherence, and academic tone throughout the manuscript. This included assistance with restructuring paragraphs, harmonizing terminology across sections, and polishing the presentation of technical concepts, experimental findings, and ablation insights.

- **Literature navigation and preliminary retrieval.** LLMs facilitated the early-stage discovery of related work in machine unlearning, contrastive decoding, and retrieval-augmented generation. These suggestions served only as initial pointers and were manually verified, curated, and cross-checked with authoritative sources before inclusion.

- **Technical drafting and scripting assistance.** LLMs provided support in drafting Python code used in various peripheral components of the project, such as preprocessing scripts, knowledge graph extraction utilities, and visualization routines. They also assisted in generating LaTeX templates for tables, figures, and appendix organization, which were subsequently adapted and finalized by the authors.

- **Formatting, organization, and workflow facilitation.** LLMs helped standardize the manuscript's formatting for Overleaf integration—e.g., recommending table layouts, caption consistency, and cross-referencing conventions. This assistance improved the overall readability and visual structure of the document without altering its scientific content.

**Input:** In which genre does Ji-Yeon Park primarily write?
**Origin Output:** Ji-Yeon Park primarily writes in the genre of **leadership**.
**CRED Output:** Ji-Yeon Park primarily writes in the **business** genre.

Figure 5: **Token Logits Maps for CRED, Origin, Retain, and Forget Prompts.** While the base model strongly activates **leadership**, CRED suppresses this forgotten concept and redirects the logits toward **business**, showing effective forgetting.

All analytical decisions, experimental configurations, data interpretations, and theoretical contributions remain entirely authored by the research team.

## H  TOKEN-LEVEL LOGIT ANALYSIS.

Inspired by (Chuang et al., 2024), Figure 5 visualizes token-level logits under four settings—Origin, Retain, Forget, and CRED. We select the top-20 tokens according to the CRED last-layer distribution and apply layer-wise min–max normalization to rescale values to the range $[0, 100]$.

Under the Origin setting, the model exhibits a clear preference for the token `leadership`, indicating that the forgotten concept remains highly activated. Incorporating retain knowledge produces only a minor shift toward `business`, and `leadership` still dominates the distribution. The Forget setting further amplifies this undesired bias.

By contrast, CRED substantially suppresses the activation of `leadership` and shifts the distribution toward `business`. This reversal shows that CRED effectively attenuates the forgotten signal while preserving the retained one, achieving stable unlearning without any retraining.

