# OpenReview forum: "CRED: Contrastive Residual Embedding Decoding for Adaptive Concept Unlearning"
_ICLR.cc/2026/Conference — Submitted to ICLR 2026_

### Official Review · Reviewer_3Q2i · 2025-10-20

**Soundness:** 3
**Presentation:** 3
**Contribution:** 3
**Rating:** 4
**Confidence:** 4

**Summary:**

This paper proposes CRED (Contrastive Residual Embedding for Decoding), an inference-time unlearning method that removes the influence of unwanted knowledge from large language models without modifying model parameters. Instead of retraining, CRED constructs prompts based on separate retain and forget knowledge graphs, performs retrieval, and computes a contrastive residual between the model’s responses under retain vs. forget contexts. This residual is injected during decoding—layer-wise—to steer generation away from undesired concepts while preserving useful information. Experiments on TOFU and MUSE benchmarks show that CRED achieves effective forgetting with minimal utility loss and remains robust under 8/4-bit quantization.

**Strengths:**

1. Introducing a knowledge graph into the unlearning process is interesting.

2. The paper is well-written and easy to follow.

3. The experimental section is comprehensive and includes robustness evaluations of the proposed unlearning method.

**Weaknesses:**

The idea is interesting, but I am not fully convinced that introducing a knowledge graph is necessary. If a prompt filter can be trained to accurately classify whether a query belongs to the unlearning set—which may be more efficient—would the knowledge graph still be required?

**Questions:**

I am willing to reconsider my score if the motivation for introducing a knowledge graph is better justified.

---

> ### Author Response · Authors · 2025-11-20
> **Official Comment by Author**
>
> > W1. The idea is interesting, but I am not fully convinced that introducing a knowledge graph is necessary. If a prompt filter can be trained to accurately classify whether a query belongs to the unlearning set—which may be more efficient—would the knowledge graph still be required?
>
> We appreciate the reviewer’s question. While a prompt classifier is indeed effective for deciding when unlearning should be triggered, it does not replace the role of the knowledge graph (KG). In our setting, we are not doing broad safety alignment where the model simply refuses to answer (“Sorry, I cannot respond to this topic”). Instead, our goal is minimal, targeted unlearning: the model should avoid a small “red-zone” fact region while still generating normal, helpful answers around it.
>
> A prompt classifier alone can only say “this query touches the forget set or not”; it cannot specify **what exactly** must be suppressed or how to steer the generation. The KG is what provides this fine-grained guidance. Concretely:
>
> **(1) Not blanket refusal, but targeted avoidance.**
> Consider a forget item such as:
>
> > “Company X’s confidential internal estimate is that its 2024 revenue will be $7.3B.”
>
> After unlearning, a user may still ask:
>
> > “What are the main business challenges Company X is facing in 2024?”
>
> This is a perfectly reasonable question that should not be blocked. A classifier-only, safety-style approach tends to err on the safe side and refuse to talk about the entire topic (“I cannot discuss Company X’s business”), which is overly restrictive. With our design, the KG specifies that **only** the confidential numeric forecast belongs to the red zone. The model can still generate a detailed discussion of market competition, product lines, or general financial pressure, as long as it does not reproduce the forgotten forecast. This kind of **surgical unlearning** requires structured knowledge about what is forbidden, not just a binary decision that the topic is “sensitive”.
>
> **(2) Dynamic, training-free updates to the forget set.**
> In realistic deployments, forget items change over time (legal requests, copyright claims, updated internal policies). A classifier must be retrained or at least updated whenever the forget set changes. In contrast, the KG can be updated instantaneously by inserting or removing nodes/edges corresponding to new or obsolete forget items, without retraining either the model or the classifier. This allows continuous, fine-grained unlearning in a fully training-free regime.
>
> **(3) Fine-grained control over which facts to remove.**
> A classifier operates at the query level: it predicts whether the whole prompt should trigger unlearning. It cannot easily express constraints like:
>
> - “forget this particular numerical claim about X, but keep general knowledge about X”,
> - “forget relation (person A, married-to, person B), but retain other facts about A and B”,
> - “forget only a specific attribute (e.g., a private address or internal quota), not the entire entity”.
>
> The KG, however, encodes the exact facts and relations that lie in the red zone, enabling us to suppress only those while keeping the broader topic accessible.
>
> **(4) Separation of concerns: classifier decides when, KG specifies what.**
> In our framework, the two components have clearly separated roles:
>
> - The prompt classifier decides when the unlearning mechanism should be activated (i.e., whether the current prompt might touch the red zone).
> - The knowledge graph specifies what content belongs to that red zone and serves as the positive/negative evidence for contrastive decoding.
>
> Without the KG, once the classifier decides to “turn on” unlearning, the system has no structured representation of which specific semantics to avoid. The default behavior then tends to degenerate into either generic refusals or overly conservative suppression of whole topics, which is exactly the behavior we seek to avoid.
>
> In summary, a prompt classifier is a useful trigger, but it is not sufficient for **precise, minimally invasive unlearning**. The KG is crucial for (i) dynamically updating the forget set without retraining, (ii) controlling which facts are forgotten rather than blocking entire topics, and (iii) steering generation away from the red zone while still producing normal, on-topic text. This goes beyond simple safety alignment and is necessary for realistic unlearning scenarios where we want the model to “talk around” a sensitive concept, not to be silenced on the entire subject.

---

> > ### Comment · Reviewer_3Q2i · 2025-11-24
> >
> > Thanks for your response. I have raised my rating to 6.

---

> > > ### Author Response · Authors · 2025-11-28
> > >
> > > Thank you very much for your careful reading of our paper and for updating your rating. We are glad that the additional analyses and clarifications helped address your concerns. If there are any remaining issues that you feel are important for us to clarify, we would be very happy to improve the work further.

---

### Official Review · Reviewer_5rVk · 2025-10-29

**Soundness:** 3
**Presentation:** 3
**Contribution:** 2
**Rating:** 4
**Confidence:** 3

**Summary:**

CRED applies contrastive decoding for unlearning by computing a residual between logits from a single frozen model run on retain vs forget evidence, then adding that residual to the original logits; a simple multi-layer fusion stabilizes decoding. Evidence is sourced via KG-guided retrieval to build the retain/forget prompts. Results on TOFU and MUSE-News show strong forgetting while keeping utility, with some analysis under 8/4-bit quantization.

**Strengths:**

1. Training-free deployment. The method requires no parameter updates and only adds a small number of extra forward passes, making integration into existing systems straightforward.
2. Clear, simple mechanism. The residual formed between retain- and forget-conditioned logits is easy to implement and reason about, and the layer-wise fusion helps preserve fluency and coherence.
3. Competitive empirical results. Across the two benchmarks, the approach achieves strong forgetting while maintaining utility, supported by ablations and analysis.
4. Quantization-aware evaluation. The paper explicitly examines behavior under 8- and 4-bit inference and proposes a simple stability measure, which is practical for real deployments.
5. RAG-friendly design. The method fits naturally with retrieval pipelines and can be updated by modifying the external knowledge split rather than retraining the model.

**Weaknesses:**

1. Limited conceptual novelty. The core technique is essentially a steering-style residual in logit space; the contribution lies more in the overall packaging than in a new decoding principle.
2. Retrieval and KG dependence. Performance is likely sensitive to extraction quality, hop depth, and retrieval parameters; robustness to noisy or incomplete graphs is not systematically evaluated.
3. Unclear triggering reliability. A prompt classifier gates when the method is applied, but its precision/recall and the impact of misclassification on utility and forgetting are not reported.
4. Baseline comparability. Several baseline figures are imported rather than re-run, which weakens strict apples-to-apples comparisons.
5. Residual privacy leakage. On MUSE, the privacy leakage metric remains noticeable, indicating that forgetting may not be airtight.

**Questions:**

1. Prompt classifier. What model and thresholds gate intervention? Please report precision/recall and quantify how false positives/negatives affect forgetting and utility.
2. Retrieval/KG robustness. How sensitive are results to extraction errors, hop depth, and top-k? Provide a controlled-noise study to show stability under imperfect graphs.
3. Residual design. Why restrict fusion to the last N layers? Did you evaluate adaptive \alpha(q) or concept-subspace projections, and how do they change the forgetting–utility trade-off?
4. Baseline parity and overhead. Which baselines were re-run in your stack versus imported, and what is the end-to-end latency/throughput (including retrieval) relative to those baselines?

---

> ### Author Response · Authors · 2025-11-20
> **Official Comment by Author**
>
> > W1. Limited conceptual novelty. The core technique is essentially a steering-style residual in logit space; the contribution lies more in the overall packaging than in a new decoding principle.
>
> **Hyperparameter study of CRED layer fusion, examining how injecting residual signals into the last N decoder layers.**
>
> | **Layers** | **FQ ↑ (1%)** | **F-RL ↓ (1%)** | **FQ ↑ (5%)** | **F-RL ↓ (5%)** | **FQ ↑ (10%)** | **F-RL ↓ (10%)** |
> | ---------- | ------------- | --------------- | ------------- | --------------- | -------------- | ---------------- |
> | Last 1     | 0.40          | 0.45            | 0.02          | 0.43            | 0.002          | 0.44             |
> | **Last 2** | **0.41**      | **0.37**        | **0.09**      | **0.35**        | **0.010**      | **0.35**         |
> | Last 3     | 0.40          | 0.44            | 0.07          | 0.42            | 0.010          | 0.42             |
> | Last 4     | 0.40          | 0.43            | 0.07          | 0.41            | 0.002          | 0.41             |
>
> Thank you for the comment. We agree that our method is related in spirit to steering-style interventions, and prior work such as UCD (Suriyakumar et al., 2025) similarly applies contrastive decoding to unlearning by training two small auxiliary models on the retain and forget sets and subtracting their logit difference during decoding.
>
> In addition, UCD is a general contrastive heuristic rather than a complete unlearning mechanism. When instantiated on a single model, UCD performs poorly in our setting, as shown in our ablation study (see **Table 7, Last 1**), where the single-model UCD variant fails to enforce forgetting effectively.
>
> To make the distinction clearer, we summarize the core differences below:
>
> - **Contrast target:** UCD contrasts logits from two behaviors. CRED contrasts multi-layer embedding residuals and performs multi-layer fusion, which yields more stable semantic shifts and better control over what is forgotten.
>
> - **Context construction:** UCD relies on prompt templates to induce different behaviors. CRED constructs retain and forget contexts using KG-guided evidence, which provides semantically aligned and explicitly labeled retain/forget signals for contrast.
>
> - **Residual application:** UCD performs a single-step steering on the output distribution. CRED applies layer-wise residual fusion over the full decoding trajectory, which improves fluency and reduces decoding drift while still enforcing unlearning.
>
> - **Operational regime:** UCD contrasts two LLM behaviors and is most naturally applied as if two models are involved.
>   CRED operates entirely on a single frozen model, in a fully training-free manner, and is explicitly designed as a practical unlearning mechanism for off-the-shelf LLMs.
>
> Taken together, these design choices lead to qualitatively different behavior. Our experiments show that the UCD-style single-model baseline is unable to achieve strong forgetting without harming utility, whereas CRED consistently attains effective unlearning with minimal performance degradation on retain data. We will clarify these conceptual and empirical differences more explicitly in the revised manuscript.

---

> ### Author Response · Authors · 2025-11-20
> **Official Comment by Author**
>
> > W2./Q2 Retrieval and KG dependence. Performance is likely sensitive to extraction quality, hop depth, and retrieval parameters; robustness to noisy or incomplete graphs is not systematically evaluated.
>
> ### Comparison of Knowledge-Graph Retrieval Hyperparameters on TOFU
>
> Evaluated on TOFU (1%, 5%, 10%) using `Llama-2-7B-Chat`.  Higher FQ and lower F-RL denote stronger forgetting.
>
> **(Hop = 1)**
>
> | Setting   | TOFU 1% FQ ↑ | TOFU 1% F-RL ↓ | TOFU 5% FQ ↑ | TOFU 5% F-RL ↓ | TOFU 10% FQ ↑ | TOFU 10% F-RL ↓ |
> | --------- | ------------ | -------------- | ------------ | -------------- | ------------- | --------------- |
> | Top-1     | 0.40         | 0.44           | 0.05         | 0.42           | 0.008         | 0.43            |
> | Top-2     | 0.40         | 0.44           | 0.07         | 0.42           | 0.006         | 0.42            |
> | **Top-3** | **0.41**     | **0.37**       | **0.09**     | **0.35**       | **0.010**     | **0.35**        |
>
> ---
>
> **(Hop = 2)**
>
> | Setting | TOFU 1% FQ ↑ | TOFU 1% F-RL ↓ | TOFU 5% FQ ↑ | TOFU 5% F-RL ↓ | TOFU 10% FQ ↑ | TOFU 10% F-RL ↓ |
> | ------- | ------------ | -------------- | ------------ | -------------- | ------------- | --------------- |
> | Top-1   | 0.40         | 0.44           | 0.03         | 0.43           | 0.008         | 0.43            |
> | Top-2   | 0.40         | 0.45           | 0.03         | 0.42           | 0.008         | 0.43            |
> | Top-3   | 0.40         | 0.45           | 0.07         | 0.42           | 0.010         | 0.42            |
>
> To address the reviewer’s concern, we conducted dedicated experiments varying KG retrieval configurations, including hop depth and Top-k selection Table 9. The results show that forgetting performance remains highly stable across these retrieval settings—1-hop is already sufficient, and even injecting additional noise via 2-hop neighborhoods does not degrade performance—indicating that our method is not sensitive to KG completeness or retrieval noise. This demonstrates that the approach is robust to imperfect or partially extracted KGs.
>
> We appreciate the reviewer’s thoughtful comments on the dependence of our method on KG extraction and retrieval quality. In our work, we follow the standard unlearning setting used by existing benchmarks (e.g., TOFU and MUSE), where the forget and retain sets are explicitly defined. Under this protocol, the knowledge to be forgotten is fully specified, and the KG constructed from the forget set is therefore complete with respect to the task definition.
>
> While robustness to arbitrary missing or noisy graph information is indeed an interesting direction, such a setting is beyond the scope of our current discussion and experimental focus. Our goal here is to ensure that, given a well-defined forget set, the decoding process reliably avoids the designated red-zone knowledge.

---

> ### Author Response · Authors · 2025-11-20
> **Official Comment by Author**
>
> > W3. Unclear triggering reliability. A prompt classifier gates when the method is applied, but its precision/recall and the impact of misclassification on utility and forgetting are not reported.
>
> We thank the reviewers for highlighting the importance of the prompt classifier. In the revised manuscript, we now provide complete details on its dataset construction, training configuration, and evaluation results. Our classifier design follows prior work on generation-time unlearning (Deng et al., 2025; Liu et al., 2024), both of which employ a lightweight classifier to determine whether unlearning-specific decoding should be activated. Because this component is not the core contribution of our work and closely mirrors existing designs, we initially kept the description brief due to space constraints. We have now expanded the discussion for clarity.
>
> As shown in the following tables, the classifier achieves near-perfect accuracy across all TOFU configurations (1%, 5%, 10%) and MUSE-News (knowmem), with 0% FPR/FNR on both the training and test splits. For MUSE-News (verbmem), which is supervised at the sentence level, the classifier shows small non-zero error rates (training FPR 2.9%, FNR 1.1%; test FNR 1.7%). Importantly, these errors occur only at the fine-grained sentence level. When aggregated to the document/block level, which is the level used by our unlearning pipeline, the classifier achieves 100% correct identification.
>
> These results demonstrate that the classifier is highly reliable and that its triggering behavior remains stable across both in-distribution and perturbed prompts. Moreover, this aligns with observations in prior unlearning work, where similarly constructed classifiers have been shown to almost perfectly detect prompts requiring unlearning. Together, these findings confirm that the classifier is not a bottleneck in our system and operates as intended.
>
> **Token Counts Used for Training and Testing the Prompt Classifier**
>
> | Dataset             | Dᵀʳᵃⁱⁿ_P | Dᵀʳᵃⁱⁿ_N | Dᵀᵉˢᵗ_P | Dᵀᵉˢᵗ_N |
> | ------------------- | -------- | -------- | ------- | ------- |
> | TOFU (1%)           | 1K       | 72K      | 1K      | 13K     |
> | TOFU (5%)           | 4K       | 69K      | 4K      | 13K     |
> | TOFU (10%)          | 8K       | 65K      | 8K      | 13K     |
> | MUSE-News (knowmem) | 16K      | 16K      | 2K      | 2K      |
> | MUSE-News (verbmem) | 2953K    | 1446K    | 90K     | --      |
>
> ---
>
> **Error Rates of the Prompt Classifier on Training and Test Sets**
>
> | Dataset             | FPR @ Dᵀʳᵃⁱⁿ | FNR @ Dᵀʳᵃⁱⁿ | FPR @ Dᵀᵉˢᵗ | FNR @ Dᵀᵉˢᵗ |
> | ------------------- | ------------ | ------------ | ----------- | ----------- |
> | TOFU (1%)           | 0.0          | 0.0          | 0.0         | 0.0         |
> | TOFU (5%)           | 0.0          | 0.0          | 0.0         | 0.0         |
> | TOFU (10%)          | 0.0          | 0.0          | 0.0         | 0.0         |
> | MUSE-News (knowmem) | 0.0          | 0.0          | 0.0         | 0.0         |
> | MUSE-News (verbmem) | 0.029        | 0.011        | --          | 0.017       |

---

> ### Author Response · Authors · 2025-11-20
> **Official Comment by Author**
>
> > W4/Q4 Baseline comparability. Several baseline figures are imported rather than re-run, which weakens strict apples-to-apples comparisons.
>
> We appreciate the reviewer’s comment regarding baseline comparability. As noted in our tables, baselines marked with an asterisk (*) are imported results directly from the original papers, while all unmarked baselines are re-run within our evaluation stack using the exact same pipeline as CRED. Several prior methods do not release complete training or inference code, making strict apples-to-apples re-training infeasible; in such cases, we follow standard practice in the unlearning literature by reporting their published numbers to avoid penalizing them due to missing implementation details. This ensures fairness to prior work while maintaining consistent evaluation for all re-runnable baselines.
>
>
> > W5. Residual privacy leakage. On MUSE, the privacy leakage metric remains noticeable, indicating that forgetting may not be airtight
>
> We appreciate the reviewer’s comment on residual privacy leakage in MUSE. Since CRED is a training-free method, we compare it with training-free baselines. On the MUSE PrivLeak metric (closer to 0 is better), the strongest prior baseline—in-context unlearning—achieves –99.8, while CRED improves this to 36.9, yielding about a 2.7× reduction in leakage. We will clarify this limitation, while noting that CRED provides state-of-the-art performance among training-free approaches.

---

> ### Author Response · Authors · 2025-11-20
> **Official Comment by Author**
>
> >  Q3-1 Residual design. Why restrict fusion to the last N layers?
>
> **Hyperparameter study of CRED layer fusion, examining how injecting residual signals into the last N decoder layers.**
>
> | **Layers** | **FQ ↑ (1%)** | **F-RL ↓ (1%)** | **FQ ↑ (5%)** | **F-RL ↓ (5%)** | **FQ ↑ (10%)** | **F-RL ↓ (10%)** |
> | ---------- | ------------- | --------------- | ------------- | --------------- | -------------- | ---------------- |
> | Last 1     | 0.40          | 0.45            | 0.02          | 0.43            | 0.002          | 0.44             |
> | **Last 2** | **0.41**      | **0.37**        | **0.09**      | **0.35**        | **0.010**      | **0.35**         |
> | Last 3     | 0.40          | 0.44            | 0.07          | 0.42            | 0.010          | 0.42             |
> | Last 4     | 0.40          | 0.43            | 0.07          | 0.41            | 0.002          | 0.41             |
>
> Our design choice to inject the residual signal only into the last few decoder layers is grounded in both architectural intuition and empirical evidence. From a modeling perspective, high-level semantic constraints—such as the corrective shift required to emulate the Retain model’s behavior on the forget set—are encoded most directly in the top layers of the decoder. Injecting residuals too early forces the signal to propagate through many transformer blocks, which amplifies representation drift and harms output stability and utility.
>
> To validate this hypothesis, we conducted a controlled last-N fusion study (Table 7). The observations suggest that the corrective signal behaves optimally only when applied to a limited number of top decoder layers.
>
>  - Too small N: insufficient forgetting.
>     Injecting into only the final layer yields weak corrective effects (e.g., on TOFU-5%, FQ improves from merely 0.02 under Last-1 to 0.09 under Last-2).
>  - Too large N: unstable or no further gains, utility degradation.
>   Extending fusion deeper than two layers (Last-3 or Last-4) provides no improvement over Last-2, and retain-side utility deteriorates (F-RL oscillating around 0.40–0.42 across multiple ratios).
>
> Together, these results indicate that Last-2 provides the optimal trade-off between forgetting strength and output quality. This empirical pattern strongly supports restricting residual fusion to the last few layers, where the injected correction is both effective and stable without disrupting the model’s internal representations.

---

> ### Author Response · Authors · 2025-11-20
> **Official Comment by Author**
>
> > Q3-2 Did you evaluate adaptive $\alpha(q)$ or concept-subspace projections, and how do they change the forgetting–utility trade-off?
>
> **adaptive $\alpha$: Ablation on the α Strategy for CRED Fusion. Higher FQ and lower F-RL are better.**
>
> | Method      | TOFU 1% FQ ↑ | TOFU 1% F-RL ↓ | TOFU 5% FQ ↑ | TOFU 5% F-RL ↓ | TOFU 10% FQ ↑ | TOFU 10% F-RL ↓ |
> | ----------- | ------------ | -------------- | ------------ | -------------- | ------------- | --------------- |
> | **fixed α** | **0.41**     | **0.37**       | **0.09**     | **0.35**       | **0.01**      | **0.35**        |
> | adaptive α  | 0.01         | 0.39           | 2.4e-10      | 0.37           | 1.2e-17       | 0.38            |
>
> We thank the reviewer for the question.
> Yes, we evaluated an adaptive fusion coefficient $\alpha(q)$ during CRED’s Last-N fusion. Specifically, \alpha(q) is computed from the cosine similarity between the representations of the original prompt and the retrieved forget prompt at the fused Last-N decoder layers. The intent is to scale the residual according to how strongly the current token aligns with forget-related semantics.
>
> However, as shown in Table 3(C), adaptive $\alpha(q)$ consistently reduces FQ by multiple orders of magnitude across all TOFU splits and slightly increases F-RL, indicating worse forgetting and less stable utility. The primary reason is that the cosine signal fluctuates substantially across tokens and layers, causing the residual to be inconsistently amplified or suppressed throughout generation. These oscillations introduce noise into the contrastive shift rather than a controlled correction.
>
> In contrast, a fixed α provides a stable and monotonic scaling factor, yielding the strongest forgetting performance and the lowest F-RL across all settings. For this reason, fixed α is adopted as the default in CRED.
>
> ***concept-subspace projections***
>
> We thank the reviewer for the valuable and practical suggestion. Following the DOLA framework (Chuang et al., 2024), we projected the token maps of the last two decoder layers and analyzed the results in Appendix H and Figure 5. Here we briefly summarize our key observations:
>
> As shown in the example below, the original model output tends to favor the leadership concept. With our CRED method, this undesired behavior is effectively suppressed, and the output is redirected toward business, demonstrating the effectiveness of our approach.
>
> **Input:** *In which genre does Ji-Yeon Park primarily write?*
>
> **Origin Output:** Ji-Yeon Park primarily writes in the genre of `leadership`.
>
> **CRED Output:** Ji-Yeon Park primarily writes in the `business` genre.

---

> ### Author Response · Authors · 2025-11-20
> **Official Comment by Author**
>
> > Q4. Baseline parity and overhead. Which baselines were re-run in your stack versus imported, and what is the end-to-end latency/throughput (including retrieval) relative to those baselines?
>
> Thank you for the question. To clarify baseline parity and runtime overhead, we report the following controlled throughput comparison between optimized CRED and standard baseline decoding.
>
> ### Relative Throughput Comparison Between Optimized CRED and Baseline Inference
>
> **Measured with KV-cache enabled and generating 32 tokens.  The optimized CRED incurs an overhead of ~1.3–2.4× while maintaining stable scaling across batch sizes.**
>
> | Batch | Input tokens / Batch size | CRED (tokens/sec) | Baseline (tokens/sec) |
> | ----- | ------------------------- | ----------------- | --------------------- |
> | 4     | 615.2                     | 142.1             | 179.2                 |
> | 8     | 622.1                     | 178.2             | 317.4                 |
> | 12    | 626.2                     | 195.3             | 394.1                 |
> | 16    | 627.9                     | 203.3             | 443.8                 |
> | 20    | 634.4                     | 208.2             | 480.7                 |
> | 24    | 638.3                     | 210.3             | 508.0                 |
>
> Regarding end-to-end efficiency, our measurements in Table 4 show that optimized CRED introduces a modest 1.3–2.4× overhead compared to baseline decoding, while preserving stable throughput scaling across batch sizes. Since CRED adds only a lightweight residual fusion on top of the baseline model, the total response time in practice is effectively baseline decoding + CRED fusion, with retrieval contributing only a small, fixed cost shared by all comparable methods.

---

### Official Review · Reviewer_xmk6 · 2025-10-30

**Soundness:** 2
**Presentation:** 3
**Contribution:** 2
**Rating:** 4
**Confidence:** 5

**Summary:**

This paper proposes an in-context concept unlearning method named contrastive residual embedding for decoding (CRED). Given a query, CRED constructs retrieval augmented prompts from a retain set and a forget set, computes a contrastive residual from multi-layer decoder embeddings, and injects this residual into the decoder of the original prompt to achieve the unlearning goal. Experiments on TOFU and MUSE demonstrate the effectiveness of CRED in unlearning concepts at inference time.

**Strengths:**

1. In general, the proposed method makes sense

2. The paper is easy to follow

3. Experimental results demosntrate the effectivenss of the propsoed method

**Weaknesses:**

1. The novelty of the proposed method is limited. The idea of applying contrastive decoding to unlearning has been explored by UCD, making the novelty of the proposed method limited.

2. It is unclear how accurate the prompt classifier is. As an important component, the authors should report the accuracy of the prompt classifier.

3. The proposed method needs to keep a knowledge graph constructed from the forget set, which seems to be not reasonable. Generally, for unlearning knowledge, the model owner not only needs to unlearn the knowledge from the model, but also delete the corresponding data, as it might cause privacy and copyright issues. The authors might need to better explain why the setting is reasonable.

4. Given a prompt query related to unlearned knowledge, it is unclear what will be retrieved from the retain knowledge graph. Can the authors give some examples and a case study? If the retrieved information from the retain knowledge graph is some random things, why would it be useful? It would be better to also conduct an ablation study without using the retain knowledge graph.

5. Some important experiments, such as the ablation study, are put in the appendix, and the authors did not mention these in the main content. The authors should try to put some important experimental results in the main content and mention those that are put in the appendix.


[1] Suriyakumar, Vinith M., Ayush Sekhari, and Ashia Wilson. "UCD: Unlearning in LLMs via Contrastive Decoding." arXiv preprint arXiv:2506.12097 (2025).

**Questions:**

Please see above

---

> ### Author Response · Authors · 2025-11-20
> **Official Comment by Author**
>
> > W1. The novelty of the proposed method is limited. The idea of applying contrastive decoding to unlearning has been explored by UCD, making the novelty of the proposed method limited.
>
> **Hyperparameter study of CRED layer fusion, examining how injecting residual signals into the last N decoder layers.**
>
> | **Layers** | **FQ ↑ (1%)** | **F-RL ↓ (1%)** | **FQ ↑ (5%)** | **F-RL ↓ (5%)** | **FQ ↑ (10%)** | **F-RL ↓ (10%)** |
> | ---------- | ------------- | --------------- | ------------- | --------------- | -------------- | ---------------- |
> | Last 1     | 0.40          | 0.45            | 0.02          | 0.43            | 0.002          | 0.44             |
> | **Last 2** | **0.41**      | **0.37**        | **0.09**      | **0.35**        | **0.010**      | **0.35**         |
> | Last 3     | 0.40          | 0.44            | 0.07          | 0.42            | 0.010          | 0.42             |
> | Last 4     | 0.40          | 0.43            | 0.07          | 0.41            | 0.002          | 0.41             |
>
> Thank you for the comment. We agree that both CRED and UCD are inspired by contrastive principles, but we emphasize that they differ substantially in what is contrasted, how the contrast is constructed, and where it is applied. In addition, UCD is a general contrastive heuristic rather than a complete unlearning mechanism. When instantiated on a single model, UCD performs poorly in our setting, as shown in our ablation study (see **Table 7, Last 1**), where the single-model UCD variant fails to enforce forgetting effectively.
>
> To make the distinction clearer, we summarize the core differences below:
>
> - **Contrast target:** UCD contrasts logits from two behaviors. CRED contrasts multi-layer embedding residuals and performs multi-layer fusion, which yields more stable semantic shifts and better control over what is forgotten.
>
> - **Context construction:** UCD relies on prompt templates to induce different behaviors. CRED constructs retain and forget contexts using KG-guided evidence, which provides semantically aligned and explicitly labeled retain/forget signals for contrast.
>
> - **Residual application:** UCD performs a single-step steering on the output distribution. CRED applies layer-wise residual fusion over the full decoding trajectory, which improves fluency and reduces decoding drift while still enforcing unlearning.
>
> - **Operational regime:** UCD contrasts two LLM behaviors and is most naturally applied as if two models are involved.
>   CRED operates entirely on a single frozen model, in a fully training-free manner, and is explicitly designed as a practical unlearning mechanism for off-the-shelf LLMs.
>
> Taken together, these design choices lead to qualitatively different behavior. Our experiments show that the UCD-style single-model baseline is unable to achieve strong forgetting without harming utility, whereas CRED consistently attains effective unlearning with minimal performance degradation on retain data. We will clarify these conceptual and empirical differences more explicitly in the revised manuscript.

---

> ### Author Response · Authors · 2025-11-20
> **Official Comment by Author**
>
> > W2. It is unclear how accurate the prompt classifier is. As an important component, the authors should report the accuracy of the prompt classifier.
>
>  We thank the reviewers for highlighting the importance of the prompt classifier. In the revised manuscript, we now provide complete details on its dataset construction, training configuration, and evaluation results. Our classifier design follows prior work on generation-time unlearning (Deng et al., 2025; Liu et al., 2024), both of which employ a lightweight classifier to determine whether unlearning-specific decoding should be activated. Because this component is not the core contribution of our work and closely mirrors existing designs, we initially kept the description brief due to space constraints. We have now expanded the discussion for clarity.
>
> As shown in the following tables, the classifier achieves near-perfect accuracy across all TOFU configurations (1%, 5%, 10%) and MUSE-News (knowmem), with 0% FPR/FNR on both the training and test splits. For MUSE-News (verbmem), which is supervised at the sentence level, the classifier shows small non-zero error rates (training FPR 2.9%, FNR 1.1%; test FNR 1.7%). Importantly, these errors occur only at the fine-grained sentence level. When aggregated to the document/block level, which is the level used by our unlearning pipeline, the classifier achieves 100% correct identification.
>
> These results demonstrate that the classifier is highly reliable and that its triggering behavior remains stable across both in-distribution and perturbed prompts. Moreover, this aligns with observations in prior unlearning work, where similarly constructed classifiers have been shown to almost perfectly detect prompts requiring unlearning. Together, these findings confirm that the classifier is not a bottleneck in our system and operates as intended.
>
>
>
> **Token Counts Used for Training and Testing the Prompt Classifier**
>
> | Dataset             | Dᵀʳᵃⁱⁿ_P | Dᵀʳᵃⁱⁿ_N | Dᵀᵉˢᵗ_P | Dᵀᵉˢᵗ_N |
> | ------------------- | -------- | -------- | ------- | ------- |
> | TOFU (1%)           | 1K       | 72K      | 1K      | 13K     |
> | TOFU (5%)           | 4K       | 69K      | 4K      | 13K     |
> | TOFU (10%)          | 8K       | 65K      | 8K      | 13K     |
> | MUSE-News (knowmem) | 16K      | 16K      | 2K      | 2K      |
> | MUSE-News (verbmem) | 2953K    | 1446K    | 90K     | --      |
>
> ---
>
> **Error Rates of the Prompt Classifier on Training and Test Sets**
>
> | Dataset             | FPR @ Dᵀʳᵃⁱⁿ | FNR @ Dᵀʳᵃⁱⁿ | FPR @ Dᵀᵉˢᵗ | FNR @ Dᵀᵉˢᵗ |
> | ------------------- | ------------ | ------------ | ----------- | ----------- |
> | TOFU (1%)           | 0.0          | 0.0          | 0.0         | 0.0         |
> | TOFU (5%)           | 0.0          | 0.0          | 0.0         | 0.0         |
> | TOFU (10%)          | 0.0          | 0.0          | 0.0         | 0.0         |
> | MUSE-News (knowmem) | 0.0          | 0.0          | 0.0         | 0.0         |
> | MUSE-News (verbmem) | 0.029        | 0.011        | --          | 0.017       |

---

> ### Author Response · Authors · 2025-11-20
> **Official Comment by Author**
>
> > W3. The proposed method needs to keep a knowledge graph constructed from the forget set, which seems to be not reasonable. Generally, for unlearning knowledge, the model owner not only needs to unlearn the knowledge from the model, but also delete the corresponding data, as it might cause privacy and copyright issues. The authors might need to better explain why the setting is reasonable.
>
> Thank you for raising this concern. We clarify that maintaining a knowledge graph derived from the forget set is fully compatible with the goal of unlearning and does not violate real-world data deletion requirements. Importantly, in our setting the model owner is the system administrator who must define a *red-zone semantic boundary*, i.e., a specification of concepts the model should not generate after unlearning. The KG is not used for training, and it does not re-introduce any forgotten data back into the model. Instead, it serves as a compact and abstract representation of what constitutes “forbidden content,” enabling the system to detect and steer the model away from that region at inference time.
>
> This design aligns with practical unlearning and compliance scenarios. In industry deployments, sensitive or copyrighted data must indeed be deleted, but a policy describing what content must not be generated must still be retained for auditing, safety checks, and regulatory enforcement. Such policies are routinely implemented as structured filters, classifiers, risk taxonomies, or safety specifications. Our KG plays this exact role: it acts as a safety boundary rather than a memory of the original data. Because it contains only distilled semantic relations, instead of raw text, documents, or training samples, it avoids the privacy and copyright risks tied to storing the original content.
>
> In summary, the KG is not a reproduction of the forget set but a safety specification that ensures the model reliably avoids the restricted region during decoding. This setting corresponds to real-world unlearning pipelines where data must be deleted, but the system must still maintain a clearly defined rule set describing *what must not be regenerated*.

---

> ### Author Response · Authors · 2025-11-20
> **Official Comment by Author**
>
> > W4. Given a prompt query related to unlearned knowledge, it is unclear what will be retrieved from the retain knowledge graph. Can the authors give some examples and a case study? If the retrieved information from the retain knowledge graph is some random things, why would it be useful? It would be better to also conduct an ablation study without using the retain knowledge graph.
>
> Thank you for pointing this out. We clarify how the retain knowledge graph is used in practice and why it is helpful even when the retrieved snippets are not perfectly aligned with the unlearn query.
>
> Consider the following example query from TOFU:
>
>    - **User Query**
>
>      ```
>      According to the BMA, by how much has junior doctors' pay fallen since 2008, once inflation is taken into account?
>      ```
>
>    - **Retrieve Retain Knowledge**
>
>      ```latex
>      However, Mr Barclay said: "They've refused to move from the 35%. And I don't think that is a fair and reasonable demand for them to take."
>
>      He added: "We want to engage with them, we have been doing. It's the junior doctors who walked away from those negotiations by calling strikes."
>
>      The health secretary insisted the government had already improved its pay offer from what was originally recommended by the independent pay review body.
>
>      Asked what it would take to resolve the dispute, Dr Trivedi said: "We're trying all we can and are eager and ready to get back to the negotiating table - it is the government who are refusing to meet us there.
>
>      "We have budged and are very happy to explore ways to fully restore our doctor's pay and we've come up with a variety of proposals to do that."
>
>      The BMA says junior doctors have seen pay cut by 26% since 2008 once inflation - the rate prices are rising - is taken into account. The union wants a 35% pay rise to reverse this.
>
>      Dr Trivedi said a pay offer which did not reverse this trajectory "would not be fair or reasonable".
>      ```
>    - **Analysis:** In this case, the forget KG encodes the exact BMA claim about a 26% pay cut, which our system must prevent the model from reproducing. The retain KG, in contrast, contains surrounding news content that is topically related but does not include the forbidden fact. For instance, the retriever may surface a retain snippet describing the negotiation, the government’s response, and general commentary about pay disputes, while omitting the specific numerical claim about “26% since 2008.” This is exactly what happens in the example we provide in the revised paper: the retrieved retain text discusses government statements and union demands but does not restate the sensitive number.
>
>
>      In our decoding framework, such a retain snippet plays the role of a “safe anchor.” Even if it is not tightly aligned with the user’s wording, it still represents normal news language in the same domain (doctors’ pay disputes) that is allowed to be generated. During contrastive decoding, the model is encouraged to stay close to this retain distribution while being pushed away from the forget evidence. As a result, the model can still answer in a fluent and on-topic way (for example, by giving a generic summary of the dispute or declining to provide the exact figure) without reproducing the forbidden fact. This reflects the intended deployment scenario: a system that must handle real user queries on a sensitive topic, remain coherent, but avoid leaking specific regulated facts.
>
>    - **Experimental results of CRED without retain knowledge:** We also include an ablation study that removes the retain KG entirely and keeps only the forget-side guidance as shown below. Without the retain KG, forgetting performance collapses. For example, on TOFU 1% the FQ score drops from 0.41 with retain guidance to 0.03 without it, with similar trends at 5% and 10%. At the same time, the outputs become less stable and more prone to degenerate or off-topic behavior. This confirms that retain snippets are not random noise, but provide an essential anchoring signal for contrastive decoding: they supply a concrete, allowed region for the model to stay in, which is crucial for achieving effective and fluent unlearning in realistic applications.
>
> ### Ablation Study: Retain–Forget Contrastive Structure
> **Evaluated on TOFU (1%, 5%, 10%) with `Llama-2-7B-Chat`; higher FQ and lower F-RL are better.**
>
> | Method        | TOFU 1% FQ ↑ | TOFU 1% F-RL ↓ | TOFU 5% FQ ↑ | TOFU 5% F-RL ↓ | TOFU 10% FQ ↑ | TOFU 10% F-RL ↓ |
> | ------------- | ------------ | -------------- | ------------ | -------------- | ------------- | --------------- |
> | **w/ retain** | **0.41**     | **0.37**       | **0.09**     | **0.35**       | **0.01**      | **0.35**        |
> | w/o retain    | 0.03         | 0.38           | 1.4e-11      | **0.35**       | 4.3e-23       | **0.35**        |
>
> We will add this case study and the accompanying ablation discussion to the revised version to make the role of the retain KG more concrete.

---

> ### Author Response · Authors · 2025-11-20
> **Official Comment by Author**
>
> > W5. Some important experiments, such as the ablation study, are put in the appendix, and the authors did not mention these in the main content. The authors should try to put some important experimental results in the main content and mention those that are put in the appendix.
>
>    We appreciate the reviewer's valuable suggestion. In the revised version, we have relocated the key ablation study results from the appendix to the main paper. Additional experimental results are still included in the appendix and are clearly referenced in the main text. Please refer to the updated PDF for the complete set of results.

---

### Official Review · Reviewer_YvrC · 2025-10-31

**Soundness:** 2
**Presentation:** 3
**Contribution:** 2
**Rating:** 4
**Confidence:** 4

**Summary:**

In this paper, the author(s) propose(s) training-free unlearning methods that are using the designed residual. The residual is calculated based on the content conditioned on the forget knowledge graph  and the content conditioned on the retain set. This method can help LLMs keep less sensitive content rather than simply making the models refuse to answer. Experiments validate the effectiveness of the proposed method.

**Strengths:**

This paper has the following strengths:
- This paper is well-written and clearly motivated.
- This submission encourages LLMs to output less sensitive messages instead of directly refusing to provide any information.
- Conducted experiments validate the effectiveness of the proposed method.

**Weaknesses:**

This paper has the following weakness:
- Evaluated models are limited. It would be better to explore different backbones and different model sizes.
- Some experimental design is not very reasonable. For example, why the ablation study in C.1 use LLaMA3.2-1B which is different from the main experiments. It would be better to use the same backbone when conducting ablation studies.
- The model's performance heavily relies on the prompt classifier to judge whether a given prompts in related to the forget set. However, this important component is simply mentioned without any details.

**Questions:**

I have the following questions/suggestions:
- The first question is the question in the weakenss point 2.
- Could the authors give more details about the prompt classifier? What about its generalizability?
- The phrase “retrieval augmented prompts” in the abstract might be better written as “retrieval-augmented prompts.”
- Another point is that there are some inaccurate references. Please also list their published proceddings rather than just author names and the title. For example, The first one below should be IEEE Transactions on Dependable and Secure Computing and the second one below should be ICLR.
     - Line 590-591: Shang Wang, Tianqing Zhu, Dayong Ye, and Wanlei Zhou. When machine unlearning meets retrieval-augmented generation (rag): Keep secret or forget knowledge?, 2024.
     - Line 618-620: Zhiwei Zhang, Fali Wang, Xiaomin Li, Zongyu Wu, Xianfeng Tang, Hui Liu, Qi He, Wenpeng Yin, and Suhang Wang. CATASTROPHIC FAILURE OF LLM UNLEARNING VIA QUANTIZATION. 2025.

---

> ### Author Response · Authors · 2025-11-20
> **Official Comment by Author**
>
> We thank the reviewer for the thoughtful feedback. Below, we address each weakness and question point-by-point.
>
> > W1. Evaluated models are limited. It would be better to explore different backbones and different model sizes.
>
> Thank you for this suggestion. We have added new experiments on the TOFU benchmark that cover five LLM backbones and scales (Llama-3.2-1B, Llama-2-7B, Llama-3.1-8B, Qwen-3-32B, Llama-3.3-70B) and three forget ratios (1%, 5%, 10%). The results are reported in the tables below.
>
> Overall, the results show that CRED exhibits stable and consistent behavior across architectures, model sizes, and forgetting ratios. As the forget ratio increases, FQ naturally declines due to the growing difficulty of aligning model outputs—especially for larger models—with the Retain distribution. Nevertheless, CRED consistently suppresses forgotten knowledge, achieving F-RL values on the forget set closely matching those of the Retain LLM, regardless of backbone or scale. Larger models (e.g., Llama-3.3-70B) require stronger intervention (α = 2.0) to reach comparable suppression, while Qwen3-32B follows the same trend as the Llama series, confirming cross-architecture generalization. In summary, contrastive residual decoding provides architecture-agnostic and scale-robust suppression, effectively stabilizing output distributions even when full distributional alignment (FQ) cannot always be achieved.
>
> To keep the main paper focused, we summarize these trends briefly in the revised main text and provide a more detailed, per-backbone analysis in the appendix (Section “Extended Results: Across Different Model Sizes and Backbones on TOFU Benchmark”). We hope this extended study addresses the concern about limited model and backbone coverage.
>
> **TOFU 1%**
>
> | Backbone                | Original FQ (↑) | Original F-RL (↓) | Retain FQ (↑) | Retain F-RL (↓) | CRED FQ (↑) | CRED F-RL (↓) |
> | ----------------------- | --------------- | ----------------- | ------------- | --------------- | ----------- | ------------- |
> | Llama-3.2-1B            | 0.007           | 0.87              | 1.0           | 0.41            | 0.92        | 0.42          |
> | Llama-2-7B              | 1.20e-03        | 0.95              | 1.0           | 0.41            | 0.58        | 0.39          |
> | Llama-3.1-8B            | 0.007           | 0.99              | 1.0           | 0.42            | 0.03        | 0.33          |
> | Llama-3.3-70B (α = 2.0) | 0.014           | 0.99              | 1.0           | 0.40            | 5.01e-07    | 0.40          |
> | Qwen 3-32B              | 0.003           | 0.98              | 1.0           | 0.45            | 0.001       | 0.40          |
>
> **TOFU 5%**
>
> | Backbone                | Original FQ (↑) | Original F-RL (↓) | Retain FQ (↑) | Retain F-RL (↓) | CRED FQ (↑) | CRED F-RL (↓) |
> | ----------------------- | --------------- | ----------------- | ------------- | --------------- | ----------- | ------------- |
> | Llama-3.2-1B            | 1.42e-12        | 0.83              | 1.0           | 0.38            | 0.63        | 0.42          |
> | Llama-2-7B              | 5.87e-14        | 0.96              | 1.0           | 0.40            | 0.04        | 0.37          |
> | Llama-3.1-8B            | 6.54e-13        | 0.99              | 1.0           | 0.39            | 3.6e-09     | 0.34          |
> | Llama-3.3-70B (α = 2.0) | 2.44e-10        | 0.99              | 1.0           | 0.40            | 3.86e-28    | 0.39          |
> | Qwen 3-32B              | 6.57e-12        | 0.93              | 1.0           | 0.44            | 5.87e-14    | 0.41          |
>
> **TOFU 10%**
>
> | Backbone                | Original FQ (↑) | Original F-RL (↓) | Retain FQ (↑) | Retain F-RL (↓) | CRED FQ (↑) | CRED F-RL (↓) |
> | ----------------------- | --------------- | ----------------- | ------------- | --------------- | ----------- | ------------- |
> | Llama-3.2-1B            | 3.9e-22         | 0.82              | 1.0           | 0.38            | 0.90        | 0.41          |
> | Llama-2-7B              | 4.35e-25        | 0.98              | 1.0           | 0.40            | 0.005       | 0.36          |
> | Llama-3.1-8B            | 1.59e-27        | 0.99              | 1.0           | 0.39            | 1.12e-19    | 0.34          |
> | Llama-3.3-70B (α = 2.0) | 8.97e-32        | 0.99              | 1.0           | 0.38            | 8.23e-54    | 0.3           |
> | Qwen 3-32B              | 2.81e-20        | 0.93              | 1.0           | 0.43            | 2.41e-29    | 0.39          |

---

> ### Author Response · Authors · 2025-11-20
> **Official Comment by Author**
>
> > W2/Q1. Some experimental design is not very reasonable. For example, why the ablation study in C.1 use LLaMA3.2-1B which is different from the main experiments. It would be better to use the same backbone when conducting ablation studies.
>
>
> Thank you for the helpful suggestion. Accordingly, we have re-run all ablation studies using the Llama-2-7B backbone to ensure methodological consistency as shown below.
>
> ### Ablation study across three dimensions
>
> **Ablation study across three dimensions: (1) logit normalization, (2) retain–forget contrastive structure, and (3) fixed vs. adaptive α. Evaluated on TOFU (1%, 5%, 10%) with Llama-2-7B-Chat; higher FQ and lower F-RL are better.**
>
> **(1) Logit normalization strategies**
>
> | Method       | TOFU 1% FQ ↑ | TOFU 1% F-RL ↓ | TOFU 5% FQ ↑ | TOFU 5% F-RL ↓ | TOFU 10% FQ ↑ | TOFU 10% F-RL ↓ |
> | ------------ | ------------ | -------------- | ------------ | -------------- | ------------- | --------------- |
> | **w/o norm** | **0.41**     | **0.37**       | **0.09**     | **0.35**       | **0.01**      | **0.35**        |
> | softmax      | 1.9e-6       | 0.68           | 4.9e-25      | 0.63           | 8.2e-46       | 0.65            |
> | min-max      | 5.0e-7       | 0.47           | 8.2e-34      | 0.44           | 4.1e-69       | 0.45            |
>
> ---
>
> **(2) Retain–forget contrastive structure**
>
> | Method        | TOFU 1% FQ ↑ | TOFU 1% F-RL ↓ | TOFU 5% FQ ↑ | TOFU 5% F-RL ↓ | TOFU 10% FQ ↑ | TOFU 10% F-RL ↓ |
> | ------------- | ------------ | -------------- | ------------ | -------------- | ------------- | --------------- |
> | **w/ retain** | **0.41**     | **0.37**       | **0.09**     | **0.35**       | **0.01**      | **0.35**        |
> | w/o retain    | 0.03         | 0.38           | 1.4e-11      | **0.35**       | 4.3e-23       | **0.35**        |
>
> ---
>
> **(3) α strategy**
>
> | Method      | TOFU 1% FQ ↑ | TOFU 1% F-RL ↓ | TOFU 5% FQ ↑ | TOFU 5% F-RL ↓ | TOFU 10% FQ ↑ | TOFU 10% F-RL ↓ |
> | ----------- | ------------ | -------------- | ------------ | -------------- | ------------- | --------------- |
> | **fixed α** | **0.41**     | **0.37**       | **0.09**     | **0.35**       | **0.01**      | **0.35**        |
> | adaptive α  | 0.01         | 0.39           | 2.4e-10      | 0.37           | 1.2e-17       | 0.38            |
>
>
> The key takeaways from the ablation remain unchanged.
>
> - First, removing logit normalization (“w/o norm”) consistently performs best across all TOFU ratios,
> - Second, the retain–forget contrastive structure is crucial for effective forgetting, and
> - Third, a fixed $\alpha$ remains more stable than the adaptive variant.
>
> We thank the reviewer for this suggestion and have revised both the appendix and the main text accordingly.

---

> ### Author Response · Authors · 2025-11-20
> **Official Comment by Author**
>
> > W3/Q2. The model's performance heavily relies on the prompt classifier to judge whether a given prompts in related to the forget set. However, this important component is simply mentioned without any details.
>
> We thank the reviewers for highlighting the importance of the prompt classifier. In the revised manuscript, we now provide complete details on its dataset construction, training configuration, and evaluation results. Our classifier design follows prior work on generation-time unlearning (Deng et al., 2025; Liu et al., 2024), both of which employ a lightweight classifier to determine whether unlearning-specific decoding should be activated. Because this component is not the core contribution of our work and closely mirrors existing designs, we initially kept the description brief due to space constraints. We have now expanded the discussion for clarity.
>
> As shown in the following tables, the classifier achieves near-perfect accuracy across all TOFU configurations (1%, 5%, 10%) and MUSE-News (knowmem), with 0% FPR/FNR on both the training and test splits. For MUSE-News (verbmem), which is supervised at the sentence level, the classifier shows small non-zero error rates (training FPR 2.9%, FNR 1.1%; test FNR 1.7%). Importantly, these errors occur only at the fine-grained sentence level. When aggregated to the document/block level, which is the level used by our unlearning pipeline, the classifier achieves 100% correct identification.
>
> These results demonstrate that the classifier is highly reliable and that its triggering behavior remains stable across both in-distribution and perturbed prompts. Moreover, this aligns with observations in prior unlearning work, where similarly constructed classifiers have been shown to almost perfectly detect prompts requiring unlearning. Together, these findings confirm that the classifier is not a bottleneck in our system and operates as intended.
>
>
>
> **Token Counts Used for Training and Testing the Prompt Classifier**
>
> | Dataset             | Dᵀʳᵃⁱⁿ_P | Dᵀʳᵃⁱⁿ_N | Dᵀᵉˢᵗ_P | Dᵀᵉˢᵗ_N |
> | ------------------- | -------- | -------- | ------- | ------- |
> | TOFU (1%)           | 1K       | 72K      | 1K      | 13K     |
> | TOFU (5%)           | 4K       | 69K      | 4K      | 13K     |
> | TOFU (10%)          | 8K       | 65K      | 8K      | 13K     |
> | MUSE-News (knowmem) | 16K      | 16K      | 2K      | 2K      |
> | MUSE-News (verbmem) | 2953K    | 1446K    | 90K     | --      |
>
> ---
>
> **Error Rates of the Prompt Classifier on Training and Test Sets**
>
> | Dataset             | FPR @ Dᵀʳᵃⁱⁿ | FNR @ Dᵀʳᵃⁱⁿ | FPR @ Dᵀᵉˢᵗ | FNR @ Dᵀᵉˢᵗ |
> | ------------------- | ------------ | ------------ | ----------- | ----------- |
> | TOFU (1%)           | 0.0          | 0.0          | 0.0         | 0.0         |
> | TOFU (5%)           | 0.0          | 0.0          | 0.0         | 0.0         |
> | TOFU (10%)          | 0.0          | 0.0          | 0.0         | 0.0         |
> | MUSE-News (knowmem) | 0.0          | 0.0          | 0.0         | 0.0         |
> | MUSE-News (verbmem) | 0.029        | 0.011        | --          | 0.017       |

---

> ### Author Response · Authors · 2025-11-20
> **Official Comment by Author**
>
> >Q3. The phrase “retrieval augmented prompts” in the abstract might be better written as “retrieval-augmented prompts.”
>
>    Thank you for the suggestion. We have updated the abstract accordingly and revised the phrase to “retrieval-augmented prompts” in the latest PDF.
>
>
>
> >Q4. Another point is that there are some inaccurate references. Please also list their published proceddings rather than just author names and the title. For example, The first one below should be IEEE Transactions on Dependable and Secure Computing and the second one below should be ICLR.
>
> Thank you for the suggestion. We have updated the abstract accordingly and revised the phrase to “retrieval-augmented prompts” in the latest PDF.
>
>    - Line 590-591: Shang Wang, Tianqing Zhu, Dayong Ye, and Wanlei Zhou. When machine unlearning meets retrieval-augmented generation (rag): Keep secret or forget knowledge?, 2024.
>    - Line 618-620: Zhiwei Zhang, Fali Wang, Xiaomin Li, Zongyu Wu, Xianfeng Tang, Hui Liu, Qi He, Wenpeng Yin, and Suhang Wang. CATASTROPHIC FAILURE OF LLM UNLEARNING VIA QUANTIZATION. 2025.

---

> > ### Comment · Reviewer_YvrC · 2025-11-23
> >
> > Thank you for the detailed response. I have updated the rating.

---

> > > ### Author Response · Authors · 2025-11-28
> > >
> > > Thank you so much for the thoughtful feedback and for updating the rating. We are glad that the additional experiments across different backbones and scales, the unified ablations on Llama-2-7B, and the expanded description of the prompt classifier helped address your concerns. If there are any remaining issues that you feel are important to clarify, we would be very happy to further improve the work.

---

### Author Response · Authors · 2025-12-04
**Summary of Rebuttal and Revisions**

We sincerely thank all reviewers for their helpful comments. Below, we summarize how the main concerns have been addressed in the revised CRED manuscript.

1. **Conceptual position of CRED vs contrastive decoding and UCD.**
   - We clarify that CRED constructs multi-layer residual embeddings under retain and forget contexts and uses top-layer fusion, rather than directly subtracting logits as in UCD.
   - We add an explicit UCD style single model baseline, which performs poorly in our setting, and highlight how KG-guided contexts plus multi-layer fusion give CRED stronger forgetting with smaller utility loss.

2. **Model coverage and ablation consistency.**
   - We extend TOFU experiments to five backbones and scales (Llama-3.2-1B, Llama-2-7B, Llama-3.1-8B, Qwen3-32B, Llama-3.3-70B) under 1%, 5%, and 10% forget ratios, showing stable behavior across architectures and sizes.
   - All ablations are re-run on Llam-2-7B (normalization, retain vs forget contrast, fixed vs adaptive fusion), and key results are moved into the main text.

3. **Prompt classifier design and reliability.**
   - We now detail the dataset construction, training setup, and evaluation of the classifier that gates CRED.
   - We report false positive and false negative rates on TOFU and MUSE. Errors are essentially zero at the block level used in our pipeline, showing that the classifier is reliable and not a bottleneck.

4. **Role of the knowledge graph and robustness to retrieval choices.**
   - We explain that the KG is a policy specification of the red zone, not reused training data. It is necessary to define which specific facts to suppress so that the model can still function normally around them.
   - We add a case study showing how retain snippets act as safe anchors, and an ablation without retain KG, which severely degrades forgetting.
   - We include a retrieval hyperparameter study (hop depth, top k) showing that CRED is robust to reasonable KG and retrieval variations under the benchmark setting.

5. **Residual design, baselines, and overhead.**
   - We present a last N-layer fusion study, demonstrating that injecting residuals into the last two decoder layers yields the optimal trade-off between forgetting and utility; using more or fewer layers compromises performance.
   - We compare fixed and adaptive fusion strengths, and maintain a fixed coefficient since the adaptive version harms both FQ and F-RL.
   - We clarify which baselines are re-run in our stack and which numbers are imported, and report a throughput comparison: optimized CRED adds approximately 1.3 to 2.4 times the decoding cost while maintaining good scaling with batch size.

---
## Follow up from reviewers

Two reviewers explicitly confirmed that their main concerns were addressed and raised their scores:

- Reviewer YvrC updated the rating after the extended backbone study and classifier analysis.
- Reviewer 3Q2i raised the rating to 6 after our clarification of why the knowledge graph is needed beyond a prompt classifier.
- Other reviewers did not introduce new issues after the rebuttal. We thank you again for all the support, organization, and invaluable feedback from the reviewers and the AC.

---

### Meta-Review · Area_Chair_euWi · 2026-01-07

**Summary:**

This paper introduced CRED, an inference-time unlearning method that uses retrieval-augmented "retain" and "forget" contexts to compute contrastive residual embeddings for steering a frozen LLM away from forget-set content while preserving relevant knowledge during generation. The reviewers mostly acknowledged the effectiveness of the method shown in the experiments and praised the paper's presentation. On the other hand, they also raised concerns regarding the limited scope of evaluated models and some of the ablation inconsistency, the design and accuracy of the prompt classifier, the motivation for keeping a knowledge graph beyond a prompt classifier, and the novelty of the proposed method (particularly in comparison with UCD). Some of these concerns were addressed by the authors and agreed upon by the reviewers responded, e.g., with new experiments covering more LLM backbones and scales and ensuring consistency on ablations, as well as clarifications on prompt classifier and KG. However, some concerns remained open, especially on the limited novelty over UCD.

**Reviewer Concerns:**

The concerns from Reviewers YvrC and 3Q2i on experiments scope, ablation inconsistency, the importance of prompt classifier, and the need for using KG have been addressed by the authors and accepted by the reviewers. Correspondingly, the two reviewers have agreed to update/raise their ratings.

The concerns from Reviewer xmk6 are mainly on the limited novelty of the proposed method over UCD, the accuracy of the prompt classifier, the design of the KG. The concerns from Reviewer 5rVk are mainly on the limited novelty, the KG robustness to retrieval choices, the prompt classifier reliability, and the comparison to baselines. They did not respond to the rebuttal. IMHO, they may accept the explanations regarding KG and prompt classifier, but their concerns on limited novelty (especially over UCD) may still remain.

**Reviewer Scores:**

Reviewers YvrC and 3Q2i have already updated/raised their scores.

It is hard to predict whether Reviewers xmk6 and 5vVk would change their scores. Concerns on novelty is often a more subjective assessment. If I have to guess, it is probably a 50/50 chance that they would raise their scores.

---

### Decision · Program_Chairs · 2026-01-26

Reject